# Unprecedented genomic diversity of RNA viruses in arthropods reveals the ancestry of negative-sense RNA viruses

Ci-Xiu Li[1,2†], Mang Shi[1,2,3†], Jun-Hua Tian[4†], Xian-Dan Lin[5†], Yan-Jun Kang[1,2†], Liang-Jun Chen[1,2], Xin-Cheng Qin[1,2], Jianguo Xu[1,2], Edward C Holmes[1,3], Yong-Zhen Zhang[1,2]*

[1]State Key Laboratory for Infectious Disease Prevention and Control, National Institute for Communicable Disease Control and Prevention, Chinese Center for Disease Control and Prevention, Beijing, China; [2]Collaborative Innovation Center for Diagnosis and Treatment of Infectious Diseases, Hangzhou, China; [3]Marie Bashir Institute for Infectious Diseases and Biosecurity, Charles Perkins Centre, School of Biological Sciences and Sydney Medical School, The University of Sydney, Sydney, Australia; [4]Wuhan Center for Disease Control and Prevention, Wuhan, China; [5]Wenzhou Center for Disease Control and Prevention, Wenzhou, China

**\*For correspondence:**
zhangyongzhen@icdc.cn

[†]These authors contributed equally to this work

**Competing interests:** The authors declare that no competing interests exist.

**Reviewing editor**: Stephen P Goff, Howard Hughes Medical Institute, Columbia University, United States

**Abstract** Although arthropods are important viral vectors, the biodiversity of arthropod viruses, as well as the role that arthropods have played in viral origins and evolution, is unclear. Through RNA sequencing of 70 arthropod species we discovered 112 novel viruses that appear to be ancestral to much of the documented genetic diversity of negative-sense RNA viruses, a number of which are also present as endogenous genomic copies. With this greatly enriched diversity we revealed that arthropods contain viruses that fall basal to major virus groups, including the vertebrate-specific arenaviruses, filoviruses, hantaviruses, influenza viruses, lyssaviruses, and paramyxoviruses. We similarly documented a remarkable diversity of genome structures in arthropod viruses, including a putative circular form, that sheds new light on the evolution of genome organization. Hence, arthropods are a major reservoir of viral genetic diversity and have likely been central to viral evolution.

## Introduction

Negative-sense RNA viruses are important pathogens that cause a variety of diseases in humans including influenza, hemorrhagic fever, encephalitis, and rabies. Taxonomically, those negative-sense RNA viruses described to date comprise at least eight virus families and four unassigned genera or species (*King et al., 2012*). Although they share (i) a homologous RNA-dependent RNA polymerase (RdRp), (ii) inverted complementary genome ends, and (iii) an encapsidated negative-sense RNA genome, these viruses display substantial diversity in terms of virion morphology and genome organization (*King et al., 2012*). One key aspect of genome organization is the number of distinct segments, which is also central to virus classification. Among negative-sense RNA viruses, the number of segments varies from one (order *Mononegavirales*; unsegmented) to two (family *Arenaviridae*), three (*Bunyaviridae*), three-to-four (*Ophioviridae*), and six-to-eight (*Orthomyxoviridae*) and is further complicated by differences in the number, structure, and arrangement of the encoded genes.

Despite their diversity and importance in infectious disease, the origins and evolutionary history of the negative-sense RNA viruses are largely obscure. Arthropods harbor a diverse range of RNA viruses, which are often divergent from those that infect vertebrates (*Marklewitz et al., 2011*, *2013*;

**eLife digest** Many illnesses, including influenza, hemorrhagic fever, and rabies, are caused by a group of viruses called negative-sense RNA viruses. The genetic information—or genome—of these viruses is encoded in strands of RNA that must be copied before they can be translated into the proteins needed to build new viruses. It is currently known that there are at least eight different families of these viruses, which have a wide range of shapes and sizes and arrange their RNA in different ways.

Insects, spiders, and other arthropods carry many different RNA viruses. Many of these viruses have not previously been studied, and those that have been studied so far are mainly those that cause diseases in humans and other vertebrates. Researchers therefore only know a limited amount about the diversity of the negative-sense RNA viruses that arthropods harbor and how these viruses evolved. Studying how viruses evolve helps scientists to understand what makes some viruses deadly and others harmless and can also help develop treatments or vaccines for the diseases caused by the viruses.

Li, Shi, Tian, Lin, Kang et al. collected 70 species of insects, spiders, centipedes, and other arthropods in China and sequenced all the negative-sense RNA viruses in the creatures. This revealed an enormous number of negative-sense RNA viruses, including 112 new viruses. Many of the newly discovered arthropod viruses appear to be the ancestors of disease-causing viruses, including influenza viruses and the filoviruses—the group that includes the Ebola virus. Indeed, it appears that arthropods host many—if not all—of the negative-sense RNA viruses that cause disease in vertebrates and plants.

While documenting the new RNA viruses and how they are related to each other, Li et al. found many different genome structures. Some genomes were segmented, which may play an important role in evolution as segments can be easily swapped to create new genetic combinations. Non-segmented and circular genomes were also found. This genetic diversity suggests that arthropods are likely to have played a key role in the evolution of new viruses by acting as a site where many different viruses can interact and exchange genetic information.

Cook et al., 2013; Ballinger et al., 2014; Qin et al., 2014; Tokarz et al., 2014a, 2014b). However, those arthropod viruses sampled to date are generally those that have a relationship with vertebrates or are known to be agents of disease (Junglen and Drosten, 2013). To determine the extent of viral diversity harbored by arthropods, as well as their evolutionary history, we performed a systematic survey of negative-sense RNA viruses using RNA sequencing (RNA-seq) on a wide range of arthropods.

## Results

### Discovery of highly divergent negative-sense RNA viruses

We focused our study of virus biodiversity and evolution on 70 potential host species from four arthropod classes: Insecta, Arachnida, Chilopoda, and Malacostraca (*Table 1* and *Figure 1*). From these samples, 16 separate cDNA libraries were constructed and sequenced, resulting in a total of 147.4 Gb of 100-base pair-end reads (*Table 1*). Blastx comparisons against protein sequences of negative-sense RNA virus revealed 108 distinct types of complete or nearly complete large (L) proteins (or polymerase protein 1 (PB1) in the case of orthomyxoviruses) that encode the relatively conserved RdRp (*Tables 2–4*). Four additional types of previously undescribed RdRp sequence (>1000 amino acids) were identified from the Transcriptome Shotgun Assembly (TSA) database. Together, these proteins exhibited an enormous diversity in terms of sequence variation and structure. Most notably, this data set of RdRp sequences is distinct from both previously described sequences and from each other, with the most divergent showing as little as 15.8% amino acid sequence identity to its closest relatives (*Tables 2–4*). Overall, these data provide evidence for at least 16 potentially new families and genera of negative-sense RNA viruses, defined as whose RdRp sequences shared less than 25% amino acid identity with existing taxa.

Next, we measured the abundance of these sequences as the number transcripts per million (TPM) within each library after the removal of rRNA reads. The abundance of viral transcripts calculated in

**Table 1**. Host and geographic information and data output for each pool of arthropod samples

| Pool | No of units | Order | Species | Locations | Data generated (bases) |
|---|---|---|---|---|---|
| Mosquitoes—Hubei | 24 | Diptera | *Aedes sp, Armigeres subalbatus, Anopheles sinensis, Culex quinquefasciatus, Culex tritaeniorhynchus* | Hubei | 26,606,799,000 |
| Mosquitoes—Zhejiang | 26 | Diptera | *Aedes albopictus, Armigeres subalbatus, Anopheles paraliae, Anopheles sinensis, Culex pipiens, Culex sp, Culex tritaeniorhynchus* | Zhejiang | 7,233,954,480 |
| True flies | 24 | Diptera | *Atherigona orientalis, Chrysomya megacephala, Lucilia sericata, Musca domestica, Sarcophaga dux, S. peregrina, S. sp* | Hubei | 6,574,954,320 |
| Horseflies | 24 | Diptera | Unidentified *Tabanidae* (5 species) | Hubei | 8,721,642,060 |
| Cockroaches | 24 | Blattodea | *Blattella germanica* | Hubei | 6,182,028,000 |
| Water striders | 12 | Hemiptera | Unidentified *Gerridae* (2 species) | Hubei | 3,154,714,200 |
| Insects mix 1 | 6 | Diptera, Coleoptera, Lepidoptera, Neuroptera | *Abraxas tenuisuffusa, Hermetia illucens,* unidentified *Chrysopidae,* unidentified *Coleoptera, Psychoda alternata,* unidentified *Diptera,* unidentified *Stratiomyidae* | Zhejiang | 7,745,172,660 |
| Insects mix 2 | 4 | Diptera, Hemiptera | Unidentified *Hippoboscidae* (2 species), *Cimex hemipterus* | Hubei | 5,916,431,520 |
| Insects mix 3 (insect near water) | 10 | Odonata, Hemiptera, Hymenoptera, Isopoda | *Pseudothemis zonata,* unidentified *Nepidae* (2 species), *Camponotus japonicus, Diplonychus sp, Asellus sp* | Hubei | 11,973,368,200 |
| Insects mix 4 (insect in the mountain) | 12 | Diptera, Orthoptera, Odonata, Hymenoptera, Hemiptera | *Psychoda alternata, Velarifictorus micado, Crocothemis servilia,* unidentified *Phoridae,* unidentified *Lampyridae, Aphelinus sp, Hyalopterus pruni, Aulacorthum magnolia* | Hubei | 6,882,491,800 |
| Ticks | 16 | Ixodida | *Dermacentor marginatus, Dermacentor sp, Haemaphysalis doenitzi, H. longicornis, H. sp, H. formosensis, Hyalomma asiaticum, Rhipicephalus microplus, Argas miniatus* | Hubei, Zhejiang, Beijing, Xinjiang | 24,708,479,580 |
| Ticks Hyalomma asiaticum | 1 | Ixodida | *Hyalomma asiaticum* | Xinjiang | 2,006,000,100 |
| Spiders | 32 | Araneae | *Neoscona nautica, Parasteatoda tepidariorum, Plexippus setipes, Pirata sp,* unidentified *Araneae* | Hubei | 11,361,912,300 |
| Shrimps | 48 | Decapoda | *Exopalaemon carinicauda, Metapenaeus sp, Solenocera crassicornis, Penaeus monodon, Litopenaeus vannamei* | Zhejiang | 5,365,359,900 |

*Table 1. Continued on next page*

*Table 1.* Continued

| Pool | No of units | Order | Species | Locations | Data generated (bases) |
|---|---|---|---|---|---|
| Crabs and barnacles | 35 | Decapoda, Scalpelliformes | *Capitulum mitella, Charybdis hellerii, C. japonica, Uca arcuata* | Zhejiang | 5,833,269,360 |
| Millipedes | 12 | Polydesmida | Unidentified *Polydesmidae* (2 species) | Hubei, Beijing | 7,176,702,400 |

this manner exhibited substantial variation (*Figure 2*, *Tables 2–4*): while the least abundant L segment (Shayang Spider Virus 3) contributed to less than 0.001% to the total non-ribosomal RNA content, the most abundant (Sanxia Water Strider Virus 1) was at a frequency of 21.2%, and up to 43.9% if we include the matching M and S segments of the virus. The remaining viral RdRp sequences fell within a range (10–1000 TPM) that matched the abundance level of highly expressed host mitochondrial genes (*Figure 2*).

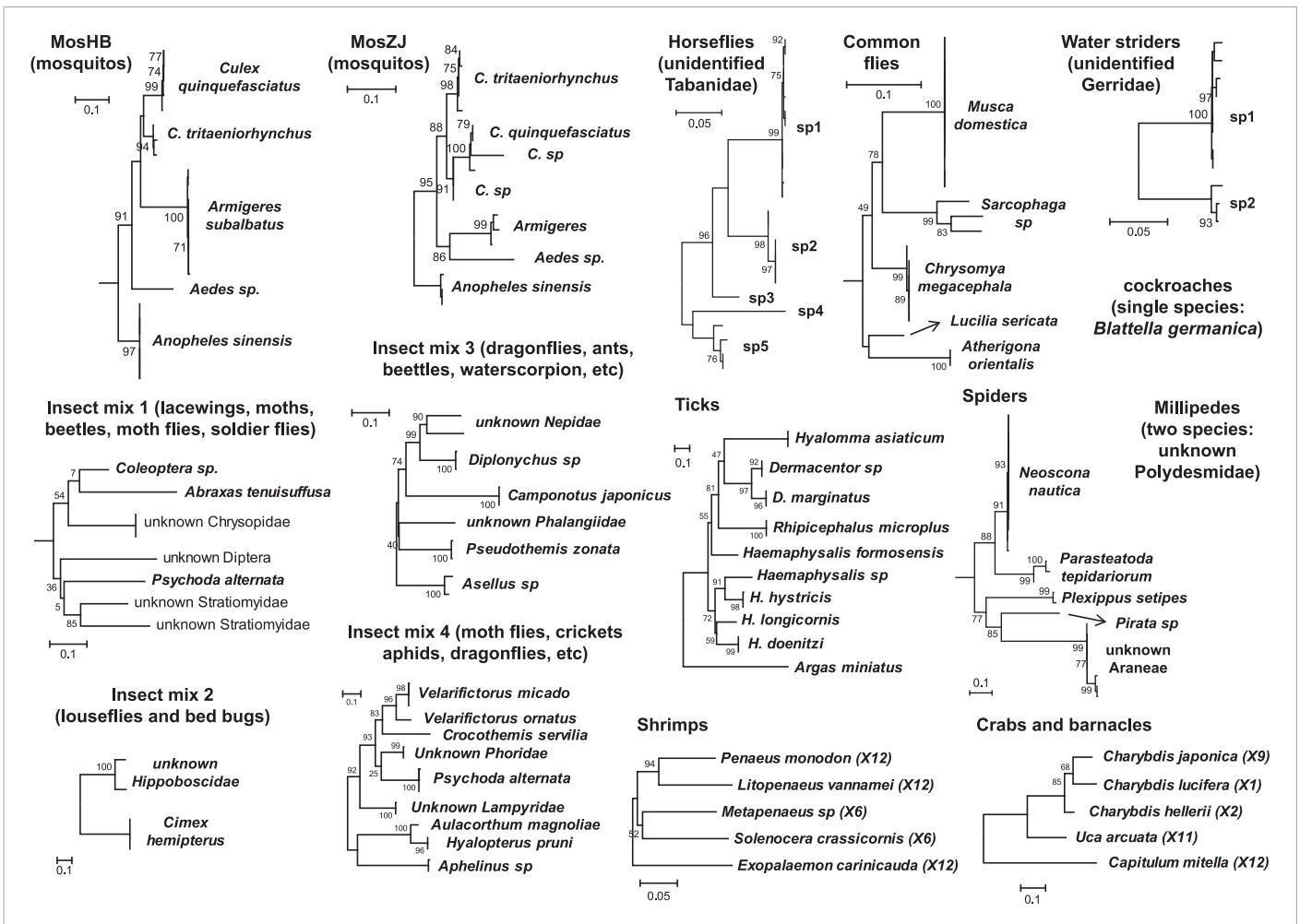

**Figure 1**. Host component of each pool used in the RNA-seq library construction and sequencing. The taxonomic units in the tree correspond to the unit samples used in the RNA extraction. Species or genus information is marked to the left of the tree.

**Table 2.** Mononegavirales-related RdRp sequences discovered in this study

| Virus name | Length of RdRp | Classification | Pool | Abundance | Putative arthropod host | Closest relative (aa identity) |
|---|---|---|---|---|---|---|
| Bole Tick Virus 3 | 2155 | Chuvirus | Ticks | 202.35 | *Hyalomma asiaticum* | Midway virus (17.1%) |
| Changping Tick Virus 2 | 2156 | Chuvirus | Ticks | 185.73 | *Dermacentor sp* | Midway virus (17.6%) |
| Changping Tick Virus 3 | 2209 | Chuvirus | Ticks | 41.80 | *Dermacentor sp* | Midway virus (16.5%) |
| Lishi Spider Virus 1 | 2180 | Chuvirus | Spiders | 5.82 | *Parasteatoda tepidariorum* | Midway virus (16.9%) |
| Shayang Fly Virus 1 | 2459 | Chuvirus | True flies | 8.99 | *Atherigona orientalis* | Maize mosaic virus (16.8%) |
| Shuangao Fly Virus 1 | 2097 | Chuvirus | Insect mix 1 | 23.63 | Unidentified *Diptera* | Lettuce big-vein associated virus (16.3%) |
| Shuangao Insect Virus 5 | 2291 | Chuvirus | Insect mix 1 | 209.31 | Unidentified *Diptera*, *Abraxas tenuisuffusa*, unidentified *Chrysopidae* | Potato yellow dwarf virus (16.3%) |
| Shuangao Lacewing Virus | 2145 | Chuvirus | Insect mix 1 | 44.48 | Unidentified *Chrysopidae* | Potato yellow dwarf virus (16.8%) |
| Tacheng Tick Virus 4 | 2101 | Chuvirus | Ticks | 137.22 | *Argas miniatus* | Midway virus (17.5%) |
| Tacheng Tick Virus 5 | 2201 | Chuvirus | Ticks | 276.32 | *Dermacentor marginatus* | Midway virus (16.8%) |
| Wenzhou Crab Virus 2 | 2208 | Chuvirus | Crabs and barnacles | 4054.25 | *Charybdis japonica*, *Charybdis lucifera*, *Charybdis hellerii* | Midway virus (15.8%) |
| Wenzhou Crab Virus 3 | 2077 | Chuvirus | Crabs and barnacles | 169.21 | *Charybdis japonica* | Midway virus (16.3%) |
| Wuchang Cockroach Virus 3 | 2203 | Chuvirus | Cockroaches | 440.14 | *Blattella germanica* | Midway virus (16.3%) |
| Wuhan Louse Fly Virus 6 | 2182 | Chuvirus | Insect mix 2 | 4.12 | Unidentified *Hippoboscidae* | Midway virus (16.4%) |
| Wuhan Louse Fly Virus 7 | 2174 | Chuvirus | Insect mix 2 | 99.83 | Unidentified *Hippoboscidae* | Midway virus (17.2%) |
| Wuhan Mosquito Virus 8 | 2159 | Chuvirus | Mosquito Hubei | 300.33 | *Culex tritaeniorhynchus, C. quinquefasciatus, Anopheles sinensis, Armigeres subalbatus* | Midway virus (16.7%) |
| Wuhan Tick Virus 2 | 2189 | Chuvirus | Ticks | 154.46 | *Rhipicephalus microplus* | Midway virus (16.7%) |
| Culex tritaeniorhynchus rhabdovirus | 2142 | Culex tritaeniorhynchus rhabdovirus | Mosquito Hubei | 3517.32 | *Culex tritaeniorhynchus, C. quinquefasciatus, Anopheles sinensis, Armigeres subalbatus, Aedes sp* | Isfahan virus (38.5%) |
| Wuhan Insect Virus 4 | 2105 | Cytorhabdovirus | Insect mix 4 | 94.92 | *Hyalopterus pruni* OR *Aphelinus sp* | Lettuce necrotic yellows virus (40.6%) |
| Wuhan Insect Virus 5 | 2098 | Cytorhabdovirus | Insect mix 4 | 622.97 | *Hyalopterus pruni* OR *Aphelinus sp* | Persimmon virus A (47.9%) |
| Wuhan Insect Virus 6 | 2079 | Cytorhabdovirus | Insect mix 4 | 991.99 | *Hyalopterus pruni* OR *Aphelinus sp* | Persimmon virus A (45.2) |
| Wuhan Louse Fly Virus 5 | 2123 | Kolente virus like | Insect mix 2 | 98.92 | Unidentified *Hippoboscidae* | Kolente virus (54.5%) |
| Yongjia Tick Virus 2 | 2113 | Nishimuro virus like | Ticks | 13.14 | *Haemaphysalis hystricis* | Nishimuro virus (54.2%) |
| Shayang Fly Virus 2 | 2170 | Sigmavirus like | True flies | 36.83 | *Musca domestica, Chrysomya megacephala* | Isfahan virus (44.1%) |
| Wuhan Fly Virus 2 | 2134 | Sigmavirus like | True flies | 18.37 | *Musca domestica, Sarcophaga sp* | Vesicular stomatitis Indiana virus (43.4%) |
| Wuhan House Fly Virus 1 | 2098 | Sigmavirus like | True flies | 31.04 | *Musca domestica* | Isfahan virus (42.8%) |
| Wuhan Louse Fly Virus 10 | 2146 | Sigmavirus like | Insect mix 2 | 235.94 | Unidentified *Hippoboscidae* | *Drosophila melanogaster* sigmavirus (51.2%) |
| Wuhan Louse Fly Virus 8 | 2145 | Sigmavirus like | Insect mix 2 | 292.11 | Unidentified *Hippoboscidae* | *Drosophila melanogaster* sigmavirus (50.6%) |

*Table 2. Continued on next page*

Table 2. Continued

| Virus name | Length of RdRp | Classification | Pool | Abundance | Putative arthropod host | Closest relative (aa identity) |
|---|---|---|---|---|---|---|
| Wuhan Louse Fly Virus 9 | 2145 | Sigmavirus like | Insect mix 2 | 69.37 | Unidentified *Hippoboscidae* | *Drosophila melanogaster* sigmavirus (51.4%) |
| Bole Tick Virus 2 | 2171 | Unclassified dimarhabdovirus 1 | Ticks | 38.19 | *Hyalomma asiaticum* | Isfahan virus (38.1%) |
| Huangpi Tick Virus 3 | 2193 | Unclassified dimarhabdovirus 1 | Ticks | 15.81 | *Haemaphysalis doenitzi* | Eel virus European X (40%) |
| Tacheng Tick Virus 3 | 2182 | Unclassified dimarhabdovirus 1 | Ticks | 96.30 | *Dermacentor marginatus* | Eel virus European X (39.8%) |
| Taishun Tick Virus | 2226 | Unclassified dimarhabdovirus 1 | Ticks | 24.56 | *Haemaphysalis hystricis* | Vesicular stomatitis Indiana virus (36.6%) |
| Wuhan Tick Virus 1 | 2191 | Unclassified dimarhabdovirus 1 | Ticks | 119.92 | *Rhipicephalus microplus* | Eel virus European X (38.3%) |
| Wuhan Insect Virus 7 | 2120 | Unclassified dimarhabdovirus 2 | Insect mix 4 | 241.7 | *Hyalopterus pruni* OR *Aphelinus sp* | Isfahan virus (42.6%) |
| Lishi Spider Virus 2 | 2201 | Unclassified mononegavirus 1 | Spiders | 5.57 | Unidentified *Araneae* | Maize fine streak virus (19.6%) |
| Sanxia Water Strider Virus 4 | 2108 | Unclassified mononegavirus 1 | Water striders | 4767.82 | Unidentified *Gerridae* | Orchid fleck virus (20.5%) |
| Tacheng Tick Virus 6 | 2068 | Unclassified mononegavirus 1 | Ticks | 17.92 | *Argas miniatus* | Maize mosaic virus (20.6%) |
| Shuangao Fly Virus 2 | 1966 | Unclassified mononegavirus 2 | Insect mix 1 | 25.94 | *Psychoda alternata* | Midway virus (21.3%) |
| Xincheng Mosquito Virus | 2026 | Unclassified mononegavirus 2 | Mosquito Hubei | 400.12 | *Anopheles sinensis* | Midway virus (19.2%) |
| Wenzhou Crab Virus 1 | 1807 | Unclassified mononegavirus 3 | Crabs and barnacles | 382.29 | *Capitulum mitella, Charybdis japonica, Charybdis lucifera* | Midway virus (22.2%) |
| Tacheng Tick Virus 7 | 2215 | Unclassified rhabdovirus 1 | Ticks | 35.86 | *Argas miniatus* | Orchid fleck virus (24.5%) |
| Jingshan Fly Virus 2 | 1970 | Unclassified rhabdovirus 2 | True flies | 4.43 | *Sarcophaga sp* | Maize fine streak virus (23.4%) |
| Sanxia Water Strider Virus 5 | 2264 | Unclassified rhabdovirus 2 | Water striders | 4373.68 | Unidentified *Gerridae* | Northern cereal mosaic virus (22.6%) |
| Shayang Fly Virus 3 | 2231 | Unclassified rhabdovirus 2 | True flies | 27.73 | *Chrysomya megacephala, Atherigona orientalis* | Maize fine streak virus (22.6%) |
| Shuangao Bedbug Virus 2 | 2207 | Unclassified rhabdovirus 2 | Insect mix 2 | 16.29 | *Cimex hemipterus* | Maize fine streak virus (22.5%) |
| Shuangao Insect Virus 6 | 2088 | Unclassified rhabdovirus 2 | Insect mix 1 | 14.37 | Unidentified *Diptera, Abraxas tenuisuffusa* | Potato yellow dwarf virus (21.2%) |
| Wuhan Ant Virus | 2118 | Unclassified rhabdovirus 2 | Insect mix 3 | 169.79 | *Camponotus japonicus* | Lettuce necrotic yellows virus (21.4%) |
| Wuhan Fly Virus 3 | 2230 | Unclassified rhabdovirus 2 | True flies | 6.00 | *Musca domestica, Sarcophaga sp* | Maize fine streak virus (21.9%) |
| Wuhan House Fly Virus 2 | 2233 | Unclassified rhabdovirus 2 | True flies | 221.04 | *Musca domestica* | Northern cereal mosaic virus (23.4%) |
| Wuhan Mosquito Virus 9 | 2260 | Unclassified rhabdovirus 2 | Mosquito Hubei | 56.19 | *Culex tritaeniorhynchus, C. quinquefasciatus, Aedes sp* | Persimmon virus A (23.2%) |
| Wuhan Louse Fly Virus 11 | 2110 | Vesiculovirus like | Insect mix 2 | 6.11 | Unidentified *Hippoboscidae* | Vesicular stomatitis Indiana virus (52.9%) |

## Evolutionary history of negative-sense RNA viruses

With this highly diverse set of RdRp sequences in hand we re-examined the evolution of all available negative-sense RNA viruses by phylogenetic analysis (*Figure 3*; *Figure 3—figure supplement 3*).

**Table 3**. Bunya-arenaviridae-related RdRp sequences discovered in this study

| Virus name | Length of RdRp | Classification | Pool | Abundance | Putative arthropod host | Closest relative (aa identity) |
|---|---|---|---|---|---|---|
| Huangpi Tick Virus 1 | 3914 | Nairovirus like | Ticks | 11.32 | *Haemaphysalis doenitzi* | Hazara virus (39.5%) |
| Tacheng Tick Virus 1 | 3962 | Nairovirus like | Ticks | 88.91 | *Dermacentor marginatus* | Hazara virus (39.6%) |
| Wenzhou Tick Virus | 3967 | Nairovirus like | Ticks | 44.30 | *Haemaphysalis hystricis* | Crimean-Congo hemorrhagic fever virus (39.1%) |
| Shayang Spider Virus 1 | 4403 | Nairovirus like | Spiders | 90.95 | *Neoscona nautica, Parasteatoda tepidariorum, Plexippus setipes* | Crimean-Congo hemorrhagic fever virus (26.2%) |
| Xinzhou Spider Virus | 4037 | Nairovirus like | Spiders | 3.79 | *Neoscona nautica, Parasteatoda tepidariorum* | Erve virus (22.9%) |
| Sanxia Water Strider Virus 1 | 3936 | Nairovirus like | Water striders | 26,483.38 | Unidentified *Gerridae* | Hazara virus (23.4%) |
| Wuhan Louse Fly Virus 1 | 2250 | Orthobunyavirus | Insect mix 2 | 67.06 | Unidentified *Hippoboscoidea* | La Crosse virus (57.8%) |
| Shuangao Insect Virus 1 | 2335 | Orthobunyavirus like | Insect mix 1 | 7.97 | Unidentified *Chrysopidae, Psychoda alternata* | Khurdun virus (29.1%) |
| Wuchang Cockroach Virus 1 | 2125 | Phasmavirus like | Cockroaches | 11,283.22 | *Blattella germanica* | Kigluaik phantom virus (35.9%) |
| GAQJ01007189 | 1554 | Phasmavirus like | Database | N/A | *Ostrinia furnacalis* | Kigluaik phantom virus (35.9%) |
| Shuangao Insect Virus 2 | 1765 | Phasmavirus like | Insect mix 1 | 36.32 | *Abraxas tenuisuffusa*, unidentified *Diptera* | Kigluaik phantom virus (31.9%) |
| Wuhan Mosquito Virus 1 | 2095 | Phasmavirus like | Mosquito Hubei, Mosquito Zhejiang | 3523.08 | *Culex tritaeniorhynchus, Anopheles sinensis, Culex quinquefasciatus* | Kigluaik phantom virus (39.5%) |
| Wuhan Mosquito Virus 2 | 2111 | Phasmavirus like | Mosquito Hubei, Mosquito Zhejiang | 39.66 | *Culex tritaeniorhynchus, Anopheles sinensis, Culex quinquefasciatus, Aedes sp* | Kigluaik phantom virus (39.6%) |
| Huangpi Tick Virus 2 | 2121 | Phlebovirus | N/A | N/A | *Haemaphysalis sp* | Uukuniemi virus (49.3%) |
| Bole Tick Virus 1 | 2148 | Phlebovirus | Ticks | 67.86 | *Hyalomma asiaticum* | Uukuniemi virus (37.9%) |
| Changping Tick Virus 1 | 2194 | Phlebovirus | Ticks | 335.25 | *Dermacentor sp* | Uukuniemi virus (39.7%) |
| Dabieshan Tick Virus | 2148 | Phlebovirus | Ticks | 250.62 | *Haemaphysalis longicornis* | Uukuniemi virus (39.2%) |
| Lihan Tick Virus | 2151 | Phlebovirus | Ticks | 60.40 | *Rhipicephalus microplus* | Uukuniemi virus (38.6%) |
| Tacheng Tick Virus 2 | 2189 | Phlebovirus | Ticks | 132.59 | *Dermacentor marginatus* | Uukuniemi virus (39.0%) |
| Yongjia Tick Virus 1 | 2138 | Phlebovirus | Ticks | 119.49 | *Haemaphysalis hystricis* | Uukuniemi virus (40.5%) |
| GAIX01000059 | 2151 | Phlebovirus like | Database | N/A | *Pararge aegeria* | Cumuto virus (24.1%) |
| GAKZ01048260 | 1583 | Phlebovirus like | Database | N/A | *Procotyla fluviatilis* | Cumuto virus (22.8%) |
| GAQJ01008681 | 2261 | Phlebovirus like | Database | N/A | *Ostrinia furnacalis* | Gouleako virus (22.0%) |
| Shuangao Insect Virus 3 | 2050 | Phlebovirus like | Insect mix 1 | 339.41 | Unidentified *Chrysopidae*, unidentified *Diptera* | Cumuto virus (23.7%) |
| Wuhan Louse Fly Virus 2 | 2327 | Phlebovirus like | Insect mix 2 | 3.57 | Unidentified *Hippoboscoidea* | Uukuniemi virus (25.2%) |
| Wuhan Insect Virus 1 | 2099 | Phlebovirus like | Insect mix 3 | 178.53 | *Asellus sp*, unidentified *Nepidae, Camponotus japonicus* | Cumuto virus (24.8%) |

*Table 3. Continued on next page*

Table 3. Continued

| Virus name | Length of RdRp | Classification | Pool | Abundance | Putative arthropod host | Closest relative (aa identity) |
|---|---|---|---|---|---|---|
| Huangshi Humpbacked Fly Virus | 2009 | Phlebovirus like | Insect mix 4 | 13.13 | Unidentified *Phoridae* | Cumuto virus (18.1%) |
| Yichang Insect Virus | 2100 | Phlebovirus like | Insect mix 4 | 71.50 | *Aulacorthum magnoliae* | Gouleako virus (45.3%) |
| Wuhan Millipede Virus 1 | 1854 | Phlebovirus like | Millipedes and insect mix 3 | 825.66 | Unidentified *Polydesmidae* | Cumuto virus (25.3%) |
| Qingnian Mosquito Virus | 2243 | Phlebovirus like | Mosquito Hubei | 17.09 | *Culex quinquefasciatus* | Razdan virus (21.0%) |
| Wutai Mosquito Virus | 2185 | Phlebovirus like | Mosquito Hubei | 70.72 | *Culex quinquefasciatus* | Rice stripe virus (26.4%) |
| Xinzhou Mosquito Virus | 2022 | Phlebovirus like | Mosquito Hubei | 98.95 | *Anopheles sinensis* | Cumuto virus (24.7%) |
| Zhee Mosquito Virus | 2443 | Phlebovirus like | Mosquito Hubei, Mosquito Zhejiang | 308.98 | *Anopheles sinensis, Armigeres subalbatus* | Cumuto virus (22.6%) |
| Wenzhou Shrimp Virus 1 | 2051 | Phlebovirus like | Shrimps | 5859.37 | *Penaeus monodon* | Uukuniemi virus (32.2%) |
| Wuhan Spider Virus | 2251 | Phlebovirus like | Spiders | 17.71 | *Neoscona nautica, Parasteatoda tepidariorum, Plexippus setipes* | Uukuniemi virus (21.7%) |
| Wuhan Fly Virus 1 | 2192 | Phlebovirus like | True flies | 68.58 | *Atherigona orientalis, Chrysomya megacephala, Sarcophaga sp, Musca domestica* | Grand Arbaud virus (27.8%) |
| Wuhan Horsefly Virus | 3117 | Tenuivirus like | Horseflies | 13.50 | Unidentified *Tabanidae* | Uukuniemi virus (28.2%) |
| Jiangxia Mosquito Virus 1 | 1889 | Unclassified segmented virus 1 | Mosquito Hubei | 11.55 | *Culex tritaeniorhynchus* | Gouleako virus (16.7%) |
| Shuangao Bedbug Virus 1 | 2015 | Unclassified segmented virus 2 | Insect mix 2 | 12.71 | *Cimex hemipterus* | Murrumbidgee virus (16.3%) |
| Jiangxia Mosquito Virus 2 | 1860 | Unclassified segmented virus 2 | Mosquito Hubei | 2.81 | *Culex tritaeniorhynchus* | Hantavirus (18.9%) |
| Shuangao Mosquito Virus | 1996 | Unclassified segmented virus 2 | Mosquito Zhejiang | 11.67 | *Armigeres subalbatus* | Hantavirus (18.7%) |
| Wenzhou Shrimp Virus 2 | 2241 | Unclassified segmented virus 3 | Shrimps | 3824.55 | *Penaeus monodon, Exopalaemon carinicauda* | La Crosse virus (19.0%) |
| Shayang Spider Virus 2 | 2165 | Unclassified segmented virus 4 | Spiders | 12.75 | *Neoscona nautica, Pirata sp, Parasteatoda tepidariorum, unidentified Araneae* | Akabane virus (16.6%) |
| Wuhan Insect Virus 2 | 2377 | Unclassified segmented virus 5 | Insect mix 4 | 223.06 | *Hyalopterus pruni* OR *Aphelinus sp* | Kigluaik phantom virus (19.2%) |
| Sanxia Water Strider Virus 2 | 2349 | Unclassified segmented virus 5 | Water striders | 707.09 | Unidentified *Gerridae* | Kigluaik phantom virus (19.8%) |
| Wuhan Millipede Virus 2 | 3709 | Unclassified segmented virus 6 | Millipedes | 1513.41 | Unidentified *Polydesmidae* | Dugbe virus (17.2%) |
| Wuhan Insect Virus 3 | 2231 | Unclassified segmented virus 7 | Insect mix 3 | 3.50 | *Asellus sp* | Herbert virus (17.2%) |

These data greatly expand the documented diversity of four viral families/orders—the *Arenaviridae*, *Bunyaviridae*, *Orthomyxoviridae*, and *Mononegavirales*—as well as of three floating genera—*Tenuivirus*, *Emaravirus*, and *Varicosavirus* (*King et al., 2012*). Most of the newly described arthropod viruses fell basal to the known genetic diversity in these taxa: their diversity either engulfed that of previously described viruses, as in the case of phlebovirus, nairovirus, and dimarhabdovirus, or appeared as novel lineages sandwiched between existing genera or families, and hence filling in a number of

**Table 4**. Orthomyxoviridae-related RdRp sequences discovered in this study

| Virus name | Length of RdRp | Classification | Pool | Abundance | Putative arthropod host | Closest relative (aa identity) |
|---|---|---|---|---|---|---|
| Jingshan Fly Virus 1 | 795 | Quaranjavirus | True flies | 21.93 | *Atherigona orientalis, Chrysomya megacephala, Sarcophaga sp, Musca domestica* | Johnston Atoll virus (36.9%) |
| Jiujie Fly Virus | 653 | Quaranjavirus | Horseflies | 10.30 | Unidentified *Tabanidae* | Johnston Atoll virus (39.7%) |
| Sanxia Water Strider Virus 3 | 789 | Quaranjavirus | Water striders | 1101.03 | Unidentified *Gerridae* | Johnston Atoll virus (36.7%) |
| Shayang Spider Virus 3 | 768 | Quaranjavirus | Spiders | 1.95 | *Neoscona nautica* | Johnston Atoll virus (38.5%) |
| Shuangao Insect Virus 4 | 793 | Quaranjavirus | Insect mix 1 | 59.90 | Unidentified *Diptera*, unidentified *Stratiomyidae* | Johnston Atoll virus (36.9%) |
| Wuhan Louse Fly Virus 3 | 784 | Quaranjavirus | Insect mix 2 | 500.77 | Unidentified *Hippoboscoidea* | Johnston Atoll virus (37.7%) |
| Wuhan Louse Fly Virus 4 | 783 | Quaranjavirus | Insect mix 2 | 96.80 | Unidentified *Hippoboscoidea* | Johnston Atoll virus (38.2%) |
| Wuhan Mosquito Virus 3 | 801 | Quaranjavirus | Mosquito Hubei | 40.07 | *Culex tritaeniorhynchus, Culex quinquefasciatus, Armigeres subalbatus* | Johnston Atoll virus (35.6%) |
| Wuhan Mosquito Virus 4 | 792 | Quaranjavirus | Mosquito Hubei | 86.21 | *Culex tritaeniorhynchus, Culex quinquefasciatus, Armigeres subalbatus* | Johnston Atoll virus (34.8%) |
| Wuhan Mosquito Virus 5 | 806 | Quaranjavirus | Mosquito Hubei | 75.05 | *Culex tritaeniorhynchus, Culex quinquefasciatus, Armigeres subalbatus* | Johnston Atoll virus (35.5%) |
| Wuhan Mosquito Virus 6 | 800 | Quaranjavirus | Mosquito Hubei | 56.30 | *Culex quinquefasciatus* | Johnston Atoll virus (34.2%) |
| Wuhan Mosquito Virus 7 | 779 | Quaranjavirus | Mosquito Hubei | 20.74 | *Anopheles sinensis, Culex quinquefasciatus* | Johnston Atoll virus (34.1%) |
| Wuhan Mothfly Virus | 710 | Quaranjavirus | Insect mix 4 | 14.47 | *Psychoda alternata* | Johnston Atoll virus (39.7%) |
| Wuchang Cockroach Virus 2 | 671 | Unclassified orthomyxovirus 1 | Cockroaches | 4.01 | *Blattella germanica* | Influenza C virus (27.0%) |

phylogenetic 'gaps' (*Figure 3*; *Figure 3—figure supplement 3*). One important example was a large monophyletic group of newly discovered viruses that fell between the major groups of segmented and unsegmented viruses (*Figure 4*); we name this putative new virus family the 'Chuviridae' reflecting the geographic location in China where most of this family were identified ('Chu' is a historical term referring to large area of China encompassing the middle and lower reaches of the Yangzi River). Also of note was that some of the previously defined families no longer appear as monophyletic. For example, although classified as distinct families, the family *Arenaviridae* fell within the genetic diversity of the family *Bunyaviridae* and as a sister group to viruses of the genus *Nairovirus*. Furthermore, the floating genus *Tenuivirus* was nested within the Phlebovirus-like clade, and another floating genus, *Emaravirus*, formed a monophyletic group with the *Orthobunyavirus* and *Tospovirus* genera (*Figure 3C*; *Figure 3—figure supplement 2*). Hence, there are important inconsistencies between the current virus classification scheme and the underlying evolutionary history of the RdRp revealed here.

A key result of this study is that much of the genetic diversity of negative-sense RNA viruses in vertebrates and plants now appears to be contained within viruses that utilize arthropods as hosts or vectors. Indeed, it is striking that all vertebrate-specific segmented and unsegmented viruses (arenavirus, bornavirus, filovirus, hantavirus, influenza viruses, lyssavirus, and paramyxovirus) fall within the genetic diversity of arthropod-associated viruses (*Figures 3, 5*). Also nested with arthropod-associated diversity were plant viruses (emaravirus, tospovirus, tenuiviruses, nucleorhabdovirus,

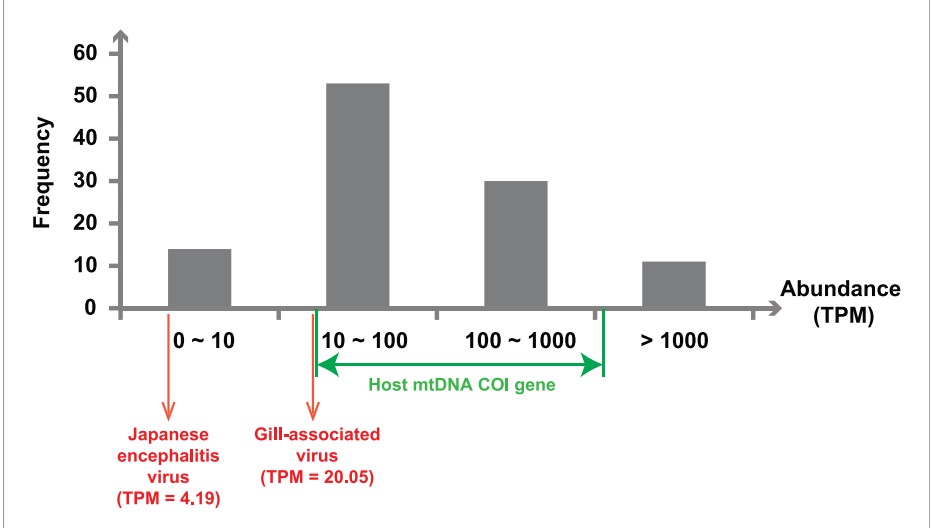

**Figure 2**. Abundance level (transcripts per million—TPM) of the RdRp genes from the negative-sense RNA viruses detected in this study. Abundance is calculated after the removal of ribosomal RNA reads. As a comparison, we show the abundance of the two well characterized (positive-sense) RNA viruses: Japanese encephalitis virus and Gill-associated virus found in the Mosquito-Hubei and Shrimp libraries, respectively, as well as the range of abundance of host mitochondrial COI genes in these same multi-host libraries.

cytorhabdovirus, and varicosavirus) (*Figures 3, 5*). Surprisingly, our phylogeny similarly placed two non-arthropod invertebrate viruses, found in nematodes (*Heterodera glycines*) and flatworms (*Procotyla fluviatilis*), within arthropod-associated diversity (*Figure 3C*, *Figure 3—figure supplement 2*), indicating that the role of non-arthropod invertebrates should be explored further. Finally, it was striking that although individual arthropod species can harbor a rich diversity of RNA viruses, many viruses seemed to be associated with different arthropod species that share the same ecological niche (*Tables 2–4*). Interestingly, host species in the same niche had similar viral contents that were generally incongruent with the host phylogeny (*Figure 6*). Such a pattern is indicative of frequent cross-species and occasional cross-genus virus transmission in the context of ecological and geographic proximity.

## Diversity and evolution of virus genome organizations

The diversity of genome structures in these virus data was also striking. This can easily be documented with respect to the evolution of genome segmentation. The number of genome segments in negative-sense RNA viruses varies from one to eight. Our phylogenetic analysis revealed no particular trend for this number to increase or decrease through evolutionary time (*Figure 4*). Hence, genome segmentation (i.e., genomes with >1 segment) has clearly evolved on multiple occasions within the negative-sense RNA viruses (*Figure 4*), such that it is a relatively flexible genetic trait. Although most segmented viruses were distantly related to those with a single segment (*Figure 4*), close phylogenetic ties were seen in other cases supporting the relatively recent evolution of multiple segments, with the plant-infecting varicosavirus (two segments) and orchid fleck virus (bipartite) serving as informative examples.

In this context, it is notable that the newly discovered chuviruses fell 'between' the phylogenetic diversity of segmented and the unsegmented viruses. Although monophyletic, the chuviruses display a wide variety of genome organizations including unsegmented, bi-segmented, and a circular form, each of which appeared multiple times in the phylogeny (*Figures 4, 7*). The circular genomic form, which was confirmed by 'around-the-genome' RT-PCR and by the mapping of sequencing reads to the genome (*Figure 7C*), is a unique feature of the Chuviridae and can be distinguished from a pseudo-circular structure seen in some other negative-sense RNA viruses including the family *Bunyaviridae* and the family *Orthomyxoviridae*. Furthermore, this circular genomic form was also present in both segments of the segmented chuviruses (*Figure 7B*). In addition, the chuviruses

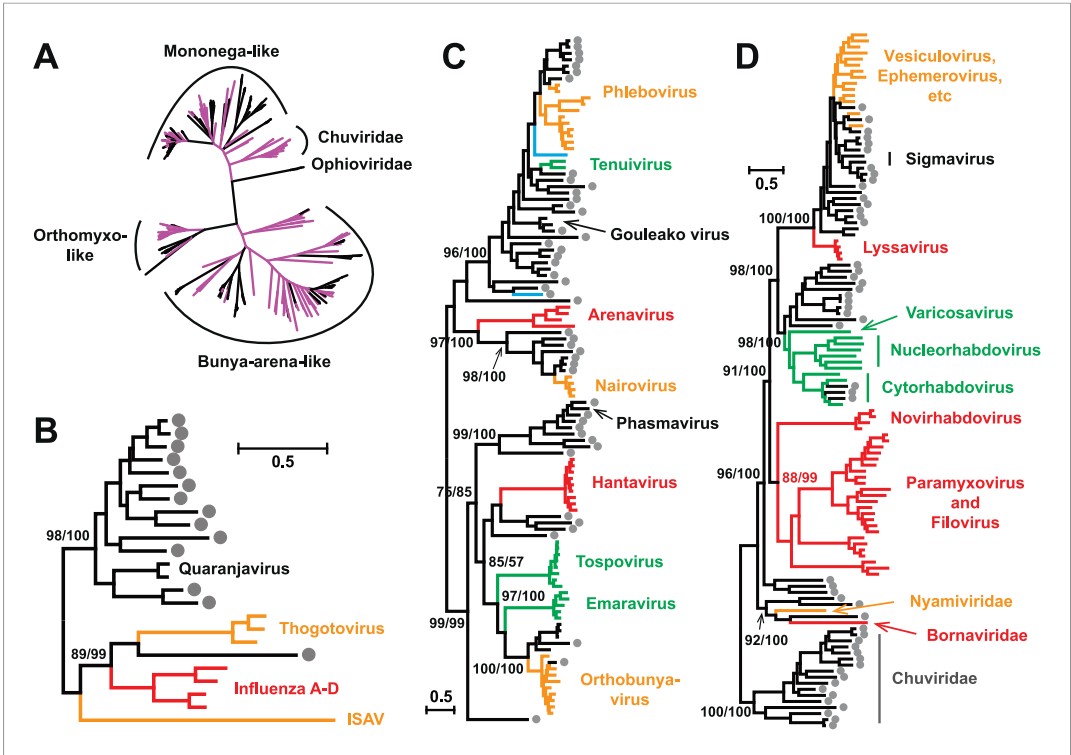

**Figure 3**. Evolutionary history of negative-sense RNA viruses based on RdRp. This is initially displayed in an unrooted maximum likelihood (ML) tree including all major groups of negative-sense RNA viruses (**A**). Separate and more detailed ML phylogenies are then shown for the Orthomyxoviridae-like (**B**), Bunya-Arenaviridae-like (**C**), and Mononegavirales-like viruses (**D**). In all the phylogenies, the RdRp sequences described here from arthropods are either shaded purple or marked with solid gray circles. The names of previously defined genera/families are labeled to the right of the phylogenies. Based on their host types, the branches are shaded red (vertebrate-specific), yellow (vertebrate and arthropod), green (plant and arthropod), blue (non-arthropod invertebrates), or black (arthropod only). For clarity, statistical supports (i.e., approximate likelihood-ratio test (aLRT) with Shimodaira–Hasegawa-like procedure/posterior probabilities) are shown for key internal nodes only.

The following figure supplements are available for figure 3:

**Figure supplement 1**. A fully labeled ML phylogeny for Orthomyxoviridae-like viruses.

**Figure supplement 2**. A fully labeled ML phylogeny for Bunya-Arenaviridae-like viruses.

**Figure supplement 3**. A fully labeled ML phylogeny for Mononegavirales-like viruses.

displayed a diverse number and arrangement of predicted open reading frames that were markedly different from the genomic arrangement seen in the order *Mononegavirales* even though these viruses are relatively closely related (*Figures 4, 7*). In particular, the chuviruses had unique and variable orders of genes: the linear chuvirus genomes began with the glycoprotein (G) gene, followed by the nucleoprotein (N) gene, and then the polymerase (L) gene, whereas the majority of circular chuviruses were most likely arranged in the order L-(G)-N (i.e., if displayed in a linear form) as the only low coverage point throughout the genome lay between the 5′ end of N gene and the 3′ end of L gene (*Figure 7B*). In addition, the genome organizations of the chuviruses were far more concise than those of the order *Mononegavirales*, with ORFs encoding only 2–3 major (>20 kDa) proteins (*Figure 7*), and hence showing more similarity to segmented viruses in this respect.

Although our phylogenetic analysis focused on the relatively conserved RdRp, in the case of segmented viruses we searched for other putative viral proteins from the assembled contigs. Accordingly, we were able to find the segments encoding matching structural proteins (mainly

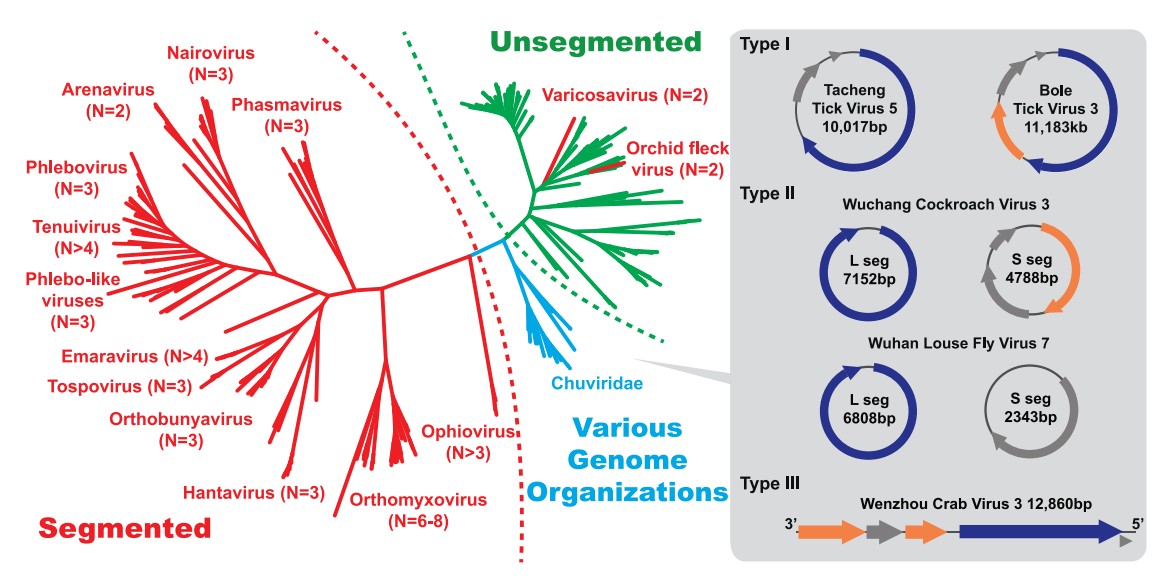

**Figure 4**. The unrooted ML phylogeny based on RdRp showing the topological position of segmented viruses within the genetic diversity of negative-sense RNA viruses. The segmented viruses are labeled with segment numbers and shaded red. The unsegmented viruses are shaded green. The Chuviridae, which exhibit a wide variety of genome organizations, are shaded cyan. Three major types of putative chuvirus genomes (circular, circular and segmented, and linear) are shown in the right panel and are annotated with predicted ORFs: putative RdRp genes are shaded blue, putative glycoprotein genes are shaded orange, and the remaining ORFs are shaded gray.

glycoproteins and nucleoproteins) for many of the viral RdRp sequences (*Figure 8*), although extensive sequence divergence prevented this in some cases. Surprisingly, M segments were apparently absent in a group of tick phleboviruses whose RdRps and nucleoproteins showed relatively high sequence similarity to Uukuniemi virus (genus *Phlebovirus*; *Table 3* and *Figure 8*). Genomes with missing glycoprotein genes were also found in the chuviruses (Changping Tick Viruses 3 and 5, Wuhan Louse Viruses 6 and 7, *Figure 7*) and the unsegmented dimarhabdovirus (Taishun Tick Virus, Wuhan Tick Virus 1, Tacheng Tick Virus 6, *Figure 9*). Although it is possible that the glycoprotein gene may have been replaced with a highly divergent or even non-homologous sequence, we failed to find any candidate G proteins within the no-Blastx-hit set of sequences under the following criteria: (i) structural resemblance to G proteins, (ii) similar level of abundance to the corresponding RdRp and nucleoprotein genes, and (iii) comparable phylogenies or levels of divergence (among related viruses) to those of RdRps and nucleoproteins. The cause and biological significance of these seemingly 'incomplete' virus genomes require further study. Finally, it was also of interest that a virus with four segments was discovered in the horsefly pool. Although the predicted proteins of all four segments showed sequence homology to their counterparts in Tenuivirus (*Falk and Tsai, 1998*), this virus lacked the ambisense coding strategy of tenuiviruses (*Figure 10*). While the capability of this virus to infect plants is unknown, it is possible that it represents a transitional form between plant-infecting and arthropod-specific viruses.

## Novel Endogenous Virus Elements (EVEs)

As well as novel exogenous RNA viruses, our metagenomic analysis also revealed a large number of potential EVEs (*Katzourakis and Gifford, 2010*) in more than 40 arthropod species; these resembled complete or partial genes of the major proteins—the nucleoprotein, glycoprotein, and RdRp—but without fully intact genomes (*Table 5*). As expected given their endogenous status, most of these sequences have disrupted reading frames and many are found within transposon elements, suggesting that transposons have been central to their integration. Interestingly, in some cases, such as the putative glycoprotein gene of chuviruses, the homologous EVEs from within a family (Culicidae) or even an order (Hymenoptera) form monophyletic groups (*Figure 11*). However, they are unlikely to be orthologous because they do not share homologous integration sites in the host genome as determined by an analysis of

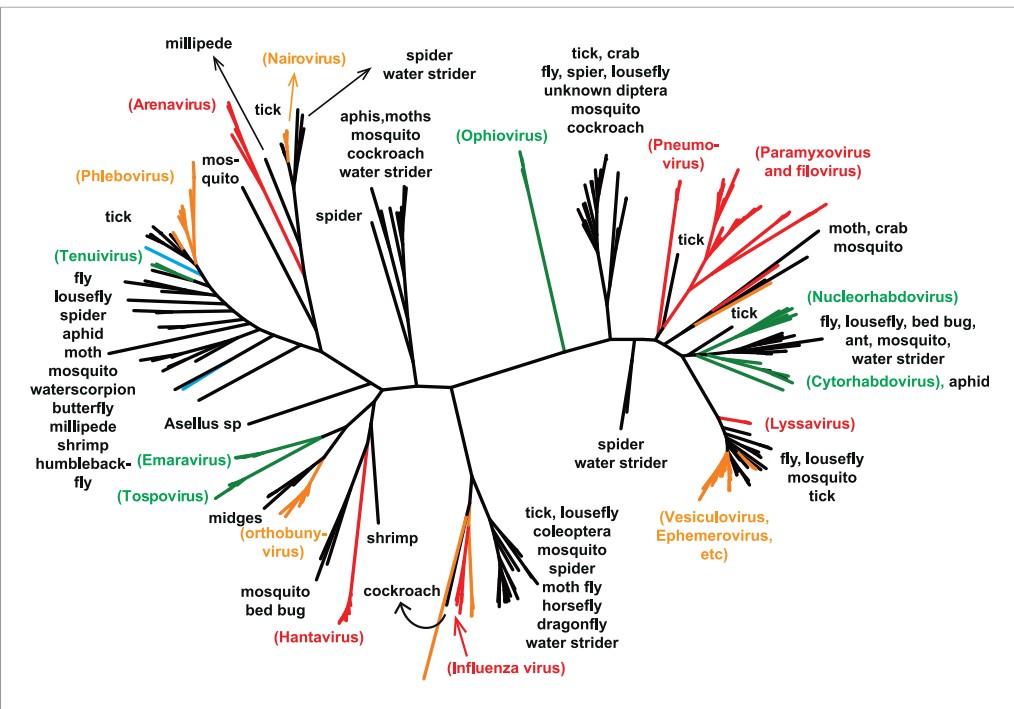

**Figure 5**. The unrooted ML phylogeny of negative-sense RNA viruses (RdRp) with the common names of the principle arthropod hosts analyzed in this study indicated. Vertebrate-specific viruses are shaded red, those infecting both vertebrates and arthropods (or with unknown vectors) are shaded yellow, those infecting both plants and arthropods are shaded green, those infecting non-arthropod invertebrates are shaded blue, and the remainder (arthropod only) are shaded black.

flanking sequences, which in turn limited the applicability of molecular-clock based dating techniques. Furthermore, phylogenetic analyses of those EVEs shared among different host species revealed extremely complex tree topologies which do not exhibit simple matches to the host phylogeny at both the species and genera levels (*Figure 11B–C*). In sum, these results suggest that EVEs are relative commonplace in arthropod genomes and have been often generated by multiple and independent integration events.

## Discussion

Our study suggests that arthropods are major reservoir hosts for many, if not all, of the negative-sense RNA viruses in vertebrates and plants, and hence have likely played a major role in their evolution. This is further supported by the high abundance of viral RNA in the arthropod transcriptome, as well as by the high frequencies of endogenous copies of these viruses in the arthropod genome, greatly expanding the known biodiversity of these genomic 'fossils' (*Katzourakis and Gifford, 2010*; *Cui and Holmes, 2012*). The often basal position of the arthropod viruses in our phylogenetic trees is also compatible with the idea that the negative-sense RNA viruses found in vertebrates and plants ultimately have their ancestry in arthropods, although this will only be confirmed with a far wider sample of virus biodiversity.

The rich genetic and phylogenetic diversity of arthropod RNA viruses may in part reflect the enormous species number and diversity of arthropods, and that they sometimes live in large and very dense populations that provide abundant hosts to fuel virus transmission. Furthermore, arthropods are involved in almost all ecological guilds and actively interact with other eukaryotes, including animals, plants, and fungi, such that it is possible that they serve as both sources and sinks for viruses present in the environment. In addition, not only were diverse viruses present, but they were often highly abundant. For example, in the pool containing 12 individuals (representing two species) from the Gerridae (Water striders) collected at the same site, we identified at least five

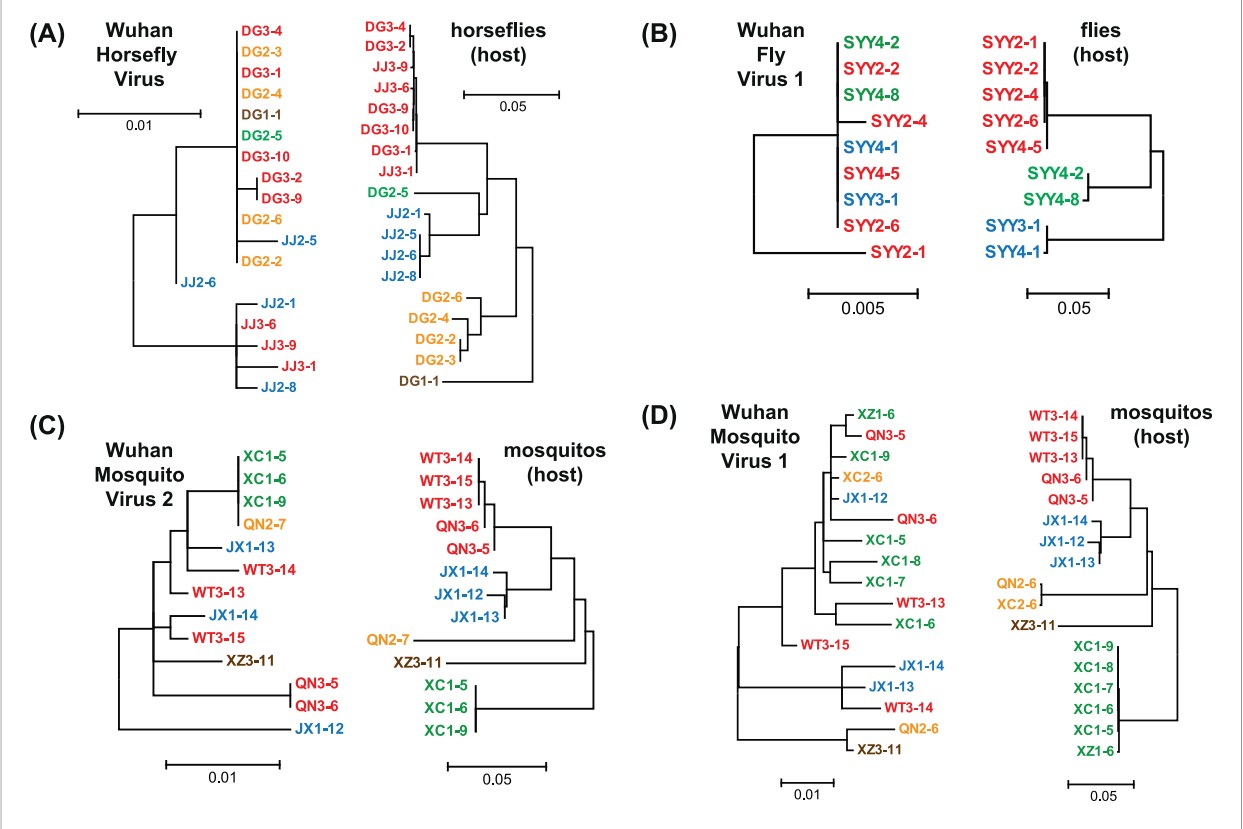

**Figure 6**. Phylogenetic congruence between viruses (M segments) and hosts. The comparisons include (**A**) Wuhan Horsefly Virus, (**B**) Wuhan Fly Virus 1, (**C**) Wuhan Mosquito Virus 2, and (**D**) Wuhan Mosquito Virus 1. Different host species/genera are distinguished with different colors, which are then mapped onto virus phylogeny to assess the phylogenetic congruence. ML phylogenetic trees were inferred in all cases.

negative-sense RNA viruses whose TPM values are well above 100, and where the viral RNA collectively made up more than 50% of the host total RNA (rRNA excluded). Determining why arthropods are able to carry such a large viral diversity and at such frequencies clearly merits further investigation.

The viruses discovered here also exhibited a huge variation in level of abundance. It is possible that this variation is in part due to the stage or severity of infection in individual viruses and may be significantly influenced by the process of pooling, since most of our libraries contain an uneven mixture of different host species or even genera. In addition, it is possible that some low abundance viruses may in fact be derived from other eukaryotic organisms present in the host sampled, such as undigested food or prey, gut micro flora, and parasites. Nevertheless, since the majority of the low abundance viruses appear in the same groups as the highly abundant ones in our phylogenetic analyses, these viruses are most likely associated with arthropods.

Viral infections in vertebrates and plants can be divided into two main categories: (i) arthropod-dependent infections, in which there is spill-over to non-arthropods but where continued virus transmission still requires arthropods, and (ii) arthropod-independent infections, in which the virus has shifted its host range to circulate among vertebrates exclusively (*Figure 12*). The first category of infections is often associated with major vector-borne diseases (*Zhang et al., 2011*, *2012*). Given the biodiversity of arthropod viruses documented here, it seems likely that arthropod-independent viruses were ultimately derived from arthropod-dependent infections, with subsequent adaptation to vertebrate-only transmission (*Figure 12*).

One of the most notable discoveries was that of a novel family, the Chuviridae. The identification of this diverse virus family provides a new perspective on the evolutionary origins of segmented and unsegmented viruses. In particular, the chuviruses occupy a phylogenetic position that is in some sense 'intermediate' between the segmented and unsegmented negative-sense RNA viruses and display

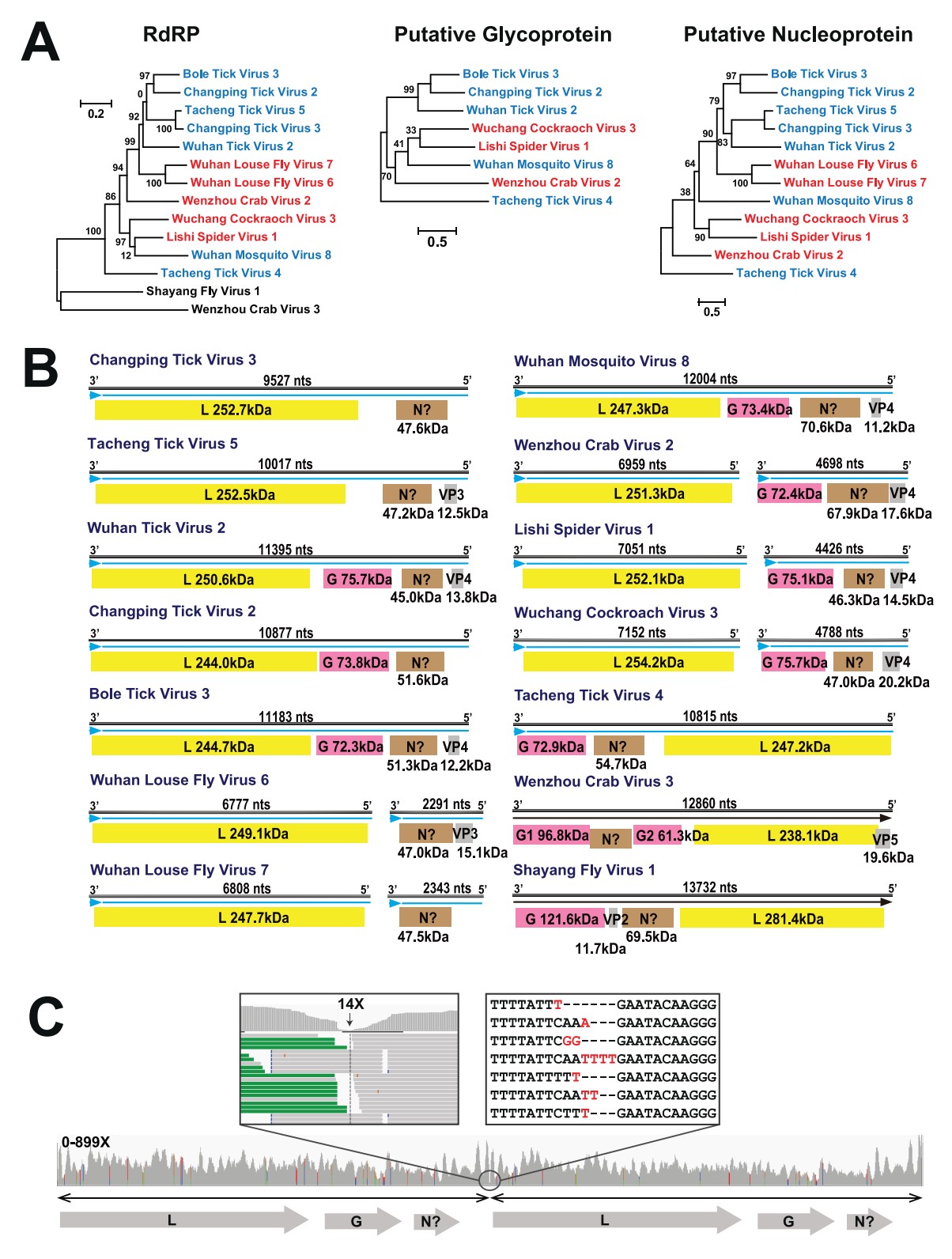

**Figure 7**. The differing genome organizations in the Chuviridae. (**A**) ML trees of three main putative proteins conserved among the chuviruses. Viruses with circular genomes (Type I) are shaded blue, while those with segmented genomes (Type II) are shaded red. (**B**) Structures of all complete chuvirus genomes. Circular genomes are indicated with the arrow (blue) situated at the 3′ end, and the genome is drawn in a linear form for ease of comparison only, being broken at the region of variable sequence (refer to the 'Materials and methods'). (**C**) An example showing mapping of sequencing reads to the

*Figure 7. continued on next page*

*Figure 7. Continued*

circular chuvirus genome. The template for mapping contains two genomes connected head-to-tail. The two boxes magnify the genomic region containing abundant sequence variation.

genomic features of both. Indeed, our phylogenetic analysis reveals that genome segmentation has evolved multiple times within the diversity of chuviruses (*Figure 7*), such that this trait appears to be more flexible than previously anticipated. In addition, the majority of the chuviruses possess circular genomes. To date, the only known circular RNA virus is (hepatitis) deltavirus, although this potentially originated from the human genome (*Salehi-Ashtiani et al., 2006*) and requires hepatitis B virus for successful replication. As such, the chuviruses may represent the first report of autonomously replicating circular RNA viruses, which opens up an important line of future research.

Our results also provide insights into the evolution of genome segmentation. Within the bunya-arena-like viruses (*Figures 3C, 4*), the three-segment structure is the most common, with the viral polymerase, nucleoprotein, and surface glycoproteins present on different segments. Notably, our phylogenetic analysis seemingly revealed independent occurrences of both increasing (Tenuivirus and Emaravirus) and decreasing (Arenavirus) segment numbers from the three-segment form (*Figure 4*). Independent changes of genome segmentation numbers are also observed in the mononegavirales-like viruses (*Figure 4*) and, more frequently, in the chuviruses (*Figure 7A*). Consequently, the number of genome segments appears to be a relatively flexible trait at a broad evolutionary scale, although

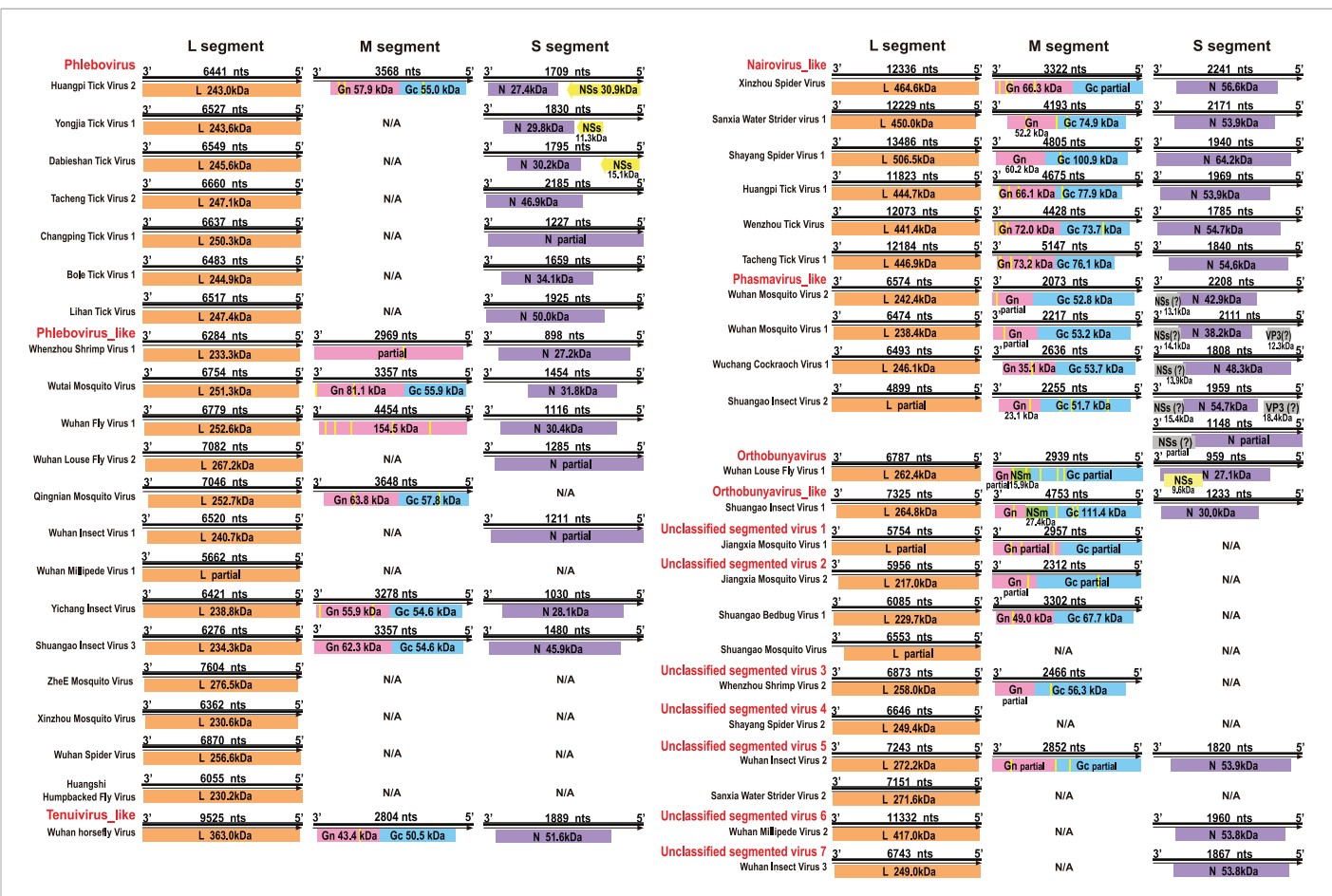

**Figure 8**. Genome structures of segmented negative-sense RNA viruses. Predicted viral proteins homologous to known viral proteins are shown and colored according to their putative functions. The numbers below each ORF box give the predicted molecular mass.

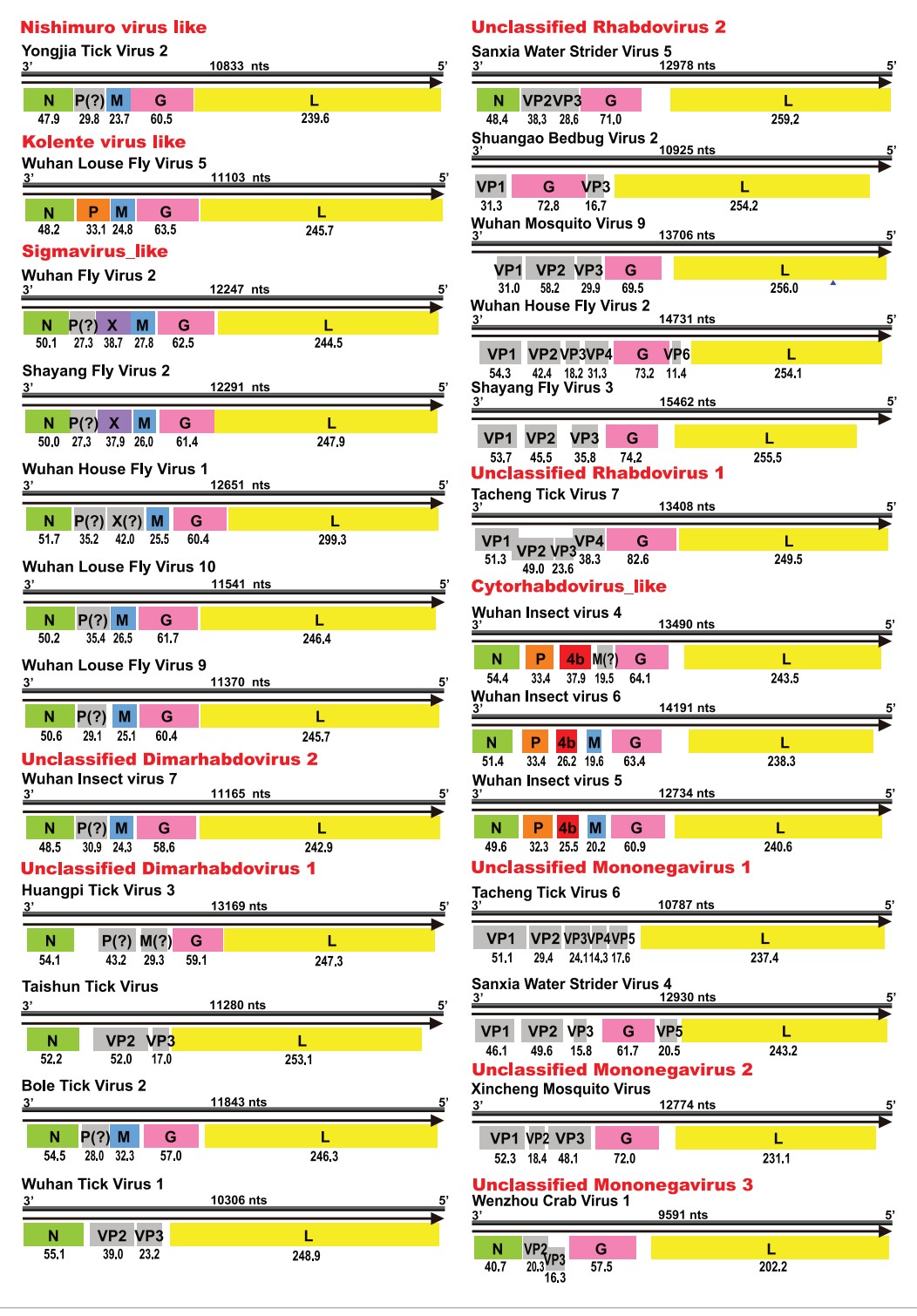

**Figure 9**. Genome structures of unsegmented negative-sense RNA viruses. Predicted ORFs encoding viral proteins with >10 kDa molecular mass are shown and colored according to their putative functions. The numbers below each ORF box give the predicted molecular mass.

the functional relevance of these changes remains unclear. While the segmented viruses (bunya-arenaviruses, orthomyxoviruses, and ophioviruses) appear to be distinct from the largely unsegmented mononegavirales-like viruses in our phylogenetic analysis, this may be an artifact of

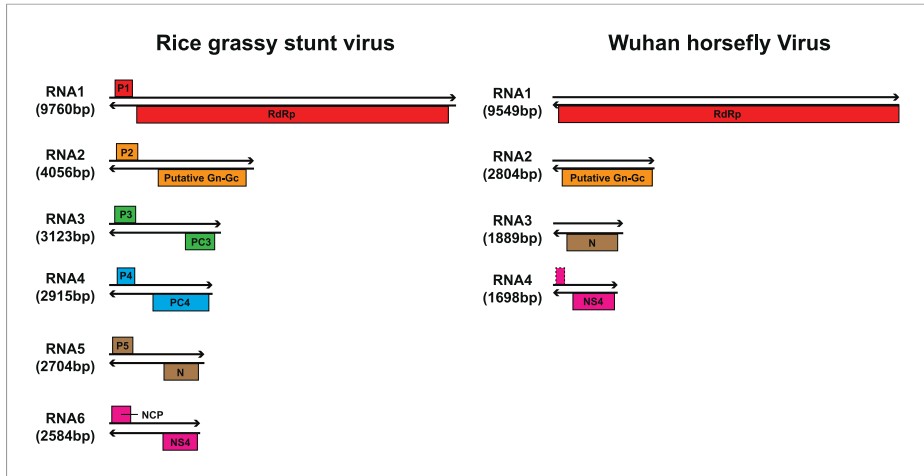

**Figure 10**. Comparison of the genome structure of a potential tenui-like virus from horsefly with a prototype tenuivirus (Rice grassy stunt virus) genome.

under-sampling, especially given that only a tiny fraction of eukaryotes have been sampled to date. With a wider sample of eukaryotic viruses it will be possible to more accurately map changes in segment number onto phylogenetic trees and in so doing come to a more complete understanding of the patterns and determinants of the evolution of genome segmentation.

In sum, our results highlight the remarkable diversity of arthropod viruses. Because arthropods interact with a wide range of organisms including vertebrate animals and plants, they can be seen as the direct or indirect source of many clinically or economically important viruses. The viral genetic and phenotypic diversity documented in arthropods here therefore provides a new perspective on fundamental questions of virus origins, diversity, host range, genome evolution, and disease emergence.

## Materials and methods

### Sample collection

Between 2011 and 2013 we collected 70 species of arthropods from various locations in China (*Table 1*). Among these, ticks were either directly picked from wild and domestic animals or captured using a tick drag-flag method; mosquitoes were trapped by light-traps; common flies were captured by fly paper; horseflies were picked from infested cattle; bed bugs and cockroaches were trapped indoors; louse flies were plucked from the skin of bats; millipedes were picked up from the ground; spiders were collected from their webs; water striders were captured using hand nets from river surfaces; and crabs and shrimps were bought (alive) from local fisherman. In addition, three pools of mixed insect samples (*Table 1*) were collected from a rural area adjacent to rice fields (Insect Mix 1), from a lakeside (Insect Mix 3), and from a mountainous area near Wuhan (Insect Mix 4). After brief species identification by experienced field biologists, these samples were immediately stored in liquid nitrogen and were later put on dry ice for shipment to our laboratory.

### Total RNA extraction

The specimens were first grouped into several units (*Table 1*). Depending on the size of specimens, one unit could include from 1 to 20 individual arthropods belonging to the same species and sampling location. These units were first washed with phosphate-buffered saline (PBS) three times before homogenized with the Mixer mill MM400 (Restsch, Germany). The resultant homogenates were then subjected to RNA extraction using TRIzol LS reagent (Invitrogen, Carlsbad, CA). After obtaining the aqueous phase containing total RNA, we performed purification steps from the E.Z.N.A Total RNA Kit (OMEGA, Portugal) according to the manufacturer's instructions. The concentration and quality of final extractions were examined using a ND-1000 UV spectrophotometer (Nanodrop, Wilmington, DE). Based on host types and/or geographic locations, these extractions were further merged into 16 pools for RNA-seq library construction and sequencing (*Table 1*).

**Table 5.** Summary of Endogenous Virus Elements (EVEs) determined here

| Host classification | Host name | Virus classification | Gene(s) present |
|---|---|---|---|
| Chelicerata | *Ixodes scapularis* | Chuvirus | G, N |
| | | Dimarhabdovirus | RdRp, N |
| | | Nairovirus like | N |
| | | Phlebovirus | RdRp, N |
| | | Quaranjavirus | RdRp |
| | *Tetranychus urticae* | Dimarhabdovirus | N |
| Crustacea | *Daphnia pulex* | Phlebovirus like | RdRp |
| | *Eurytemora affinis* | Chuvirus | G |
| | | Dimarhabdovirus | RdRp, N |
| | *Hyalella azteca* | Chuvirus | G, N |
| | | Unclassified mononegavirus 3 | RdRp, N |
| | *Lepeophtheirus salmonis* | Phlebovirus like | N, G |
| Insecta: Coleoptera | *Dendroctonus ponderosae* | Chuvirus | G |
| | | Phasmavirus | G, N |
| | *Tribolium castaneum* | Chuvirus | G |
| Insecta: Diptera | *Aedes aegypti* | Chuvirus | RdRp |
| | | Dimarhabdovirus | RdRp, N |
| | | Phasmavirus | G |
| | | Phlebovirus like | N |
| | | Quaranjavirus | RdRp |
| | *Anopheles spp.* | Chuvirus | G |
| | | Dimarhabdovirus | RdRp, N |
| | | Phasmavirus | G, N |
| | | Phlebovirus like | N |
| | | Quaranjavirus | RdRp |
| | *Culex quinquefasciatus* | Chuvirus | G, N |
| | | Dimarhabdovirus | N |
| | *Drosophila spp.* | Dimarhabdovirus | RdRp, N |
| | | Phasmavirus | N |
| | | Unclassified rhabdovirus 2 | RdRp, N |
| Insecta: Isoptera | *Zootermopsis nevadensis* | Chuvirus | N |
| Insecta: Hemiptera | *Acyrthosiphon pisum* | Chuvirus | G, N |
| | | Dimarhabdovirus | N |
| | | Phlebovirus like | N |
| | | Quaranjavirus | RdRp |
| | | Unclassified mononegavirus 1 | RdRp, N |
| | *Rhodnius prolixus* | Chuvirus | G |
| | | Phasmavirus | G |
| Insecta: Hymenoptera | *Atta cephalotes* | Unclassified mononegavirus 2 | RdRp |
| | *Acromyrmex echinatior* | Chuvirus | G |
| | | Unclassified mononegavirus 2 | RdRp |
| | *Camponotus floridanus* | Chuvirus | G |
| | | Unclassified mononegavirus 1 | N |
| | | Unclassified mononegavirus 3 | RdRp |
| | | Unclassified rhabdovirus 2 | RdRp |
| | *Harpegnathos saltator* | Chuvirus | G |
| | *Linepithema humile* | Chuvirus | G |
| | *Nasonia spp.* | Chuvirus | G |
| | *Pogonomyrmex barbatus* | Chuvirus | G |
| | *Solenopsis invicta* | Chuvirus | G |
| | | Unclassified mononegavirus 1 | N |
| | | Unclassified mononegavirus 3 | RdRp, N |

*Table 5. Continued on next page*

*Table 5. Continued*

| Host classification | Host name | Virus classification | Gene(s) present |
|---|---|---|---|
| Insecta: Lepidoptera | *Bombyx mori* | Chuvirus | RdRp, G |
| | | Quaranjavirus | RdRp |
| | | Unclassified rhabdovirus 2 | RdRp |
| | *Melitaea cinxia* | Dimarhabdovirus | N |
| | | Quaranjavirus | RdRp |
| | *Plutella xylostella* | Dimarhabdovirus | N, G |
| | *Spodoptera frugiperda* | Phlebovirus like | G |
| Myriapoda | *Strigamia maritima* | Chuvirus | N |
| | | Phlebovirus like | G |

## Species identification

To verify the field species identification, we took a proportion of the homogenates from each specimen or specimen pool for genomic DNA extraction using E.Z.N.A. DNA/RNA Isolation Kit (OMEGA). Two genes were used for host identification: the partial 18S rRNA gene (~1100 nt) which was amplified using primer pairs 18S#1 (5′-CTGGTGCCAGCGAGCCGCGGYAA-3′) and 18S#2RC (5′-TCCGTCAATTYCTTTAAGTT-3′) and partial COI gene (~680 nt) using primer pairs LCO1490 (5′-GGTCAACAAATCATAAAGATATTGG-3′) and HCO2198 (5′-TAAACTTCAGGGTGACCAAAAAATCA-3′). PCRs were performed as described previously (*Folmer et al., 1994*; *Machida and Knowlton, 2012*). For taxonomic determination, the resulting sequences were compared against the nt database as well as with all COI barcode records on the Barcode of Life Data Systems (BOLD).

## RNA-seq sequencing and reads assembly

Total RNA was subjected to a slightly modified RNA-seq library preparation protocol from that provided by Illumina. Briefly, following DNase I digestion, total RNA was subjected to an rRNA removal step using Ribo-Zero Magnetic Gold Kit (Epicentre, Madison, WI). The remaining RNA was then fragmented, reverse-transcribed, ends repaired, dA-tailed, adaptor ligated, purified, and quantified with Agilent 2100 Bioanalyzer and ABI StepOnePlus Real-Time PCR System. Pair-end (90 bp or 100 bp) sequencing of the RNA library was performed on the HiSeq 2000 platform (Illumina, San diego, CA). All library preparation and sequencing steps were performed by BGI Tech (Shenzhen, China). The resulting sequencing reads were quality trimmed and assembled de novo using the Trinity program (*Grabherr et al., 2011*). All sequence reads generated in this study were uploaded onto NCBI Sequence Read Achieve (SRA) database under the BioProject accession SRP051790.

## Discovery of target virus sequences

The assembled contigs were translated and compared (using Blastx) to reference protein sequences of all negative-sense RNA viruses. Sequences yielding e-values larger than $1e^{-5}$ were retained and compared to the entire nr database to exclude non-viral sequences. The resulting viral sequences were merged by identifying unassembled overlaps between neighboring contigs or within a scaffold using the SeqMan program implemented in the Lasergene software package v7.1 (DNAstar, Madison, WI). To prevent missing highly divergent viruses, the newly found viral sequences were included in the reference protein sequences for a second round of Blastx.

## Sequence confirmation and repairing by Sanger methods

For each potential viral sequence, we first used nested RT-PCR to examine which unit contained the target sequence, utilizing primers designed based on the deep-sequencing results. In the case of segmented viruses this information was also used to determine whether and which of the segments recovered from the pool belonged to the same virus. We next designed overlapping primers to verify the sequence obtained from the deep sequencing and assembly processes. Based on the verified sequences, we determined the sequencing depth and coverage by mapping reads to target sequences

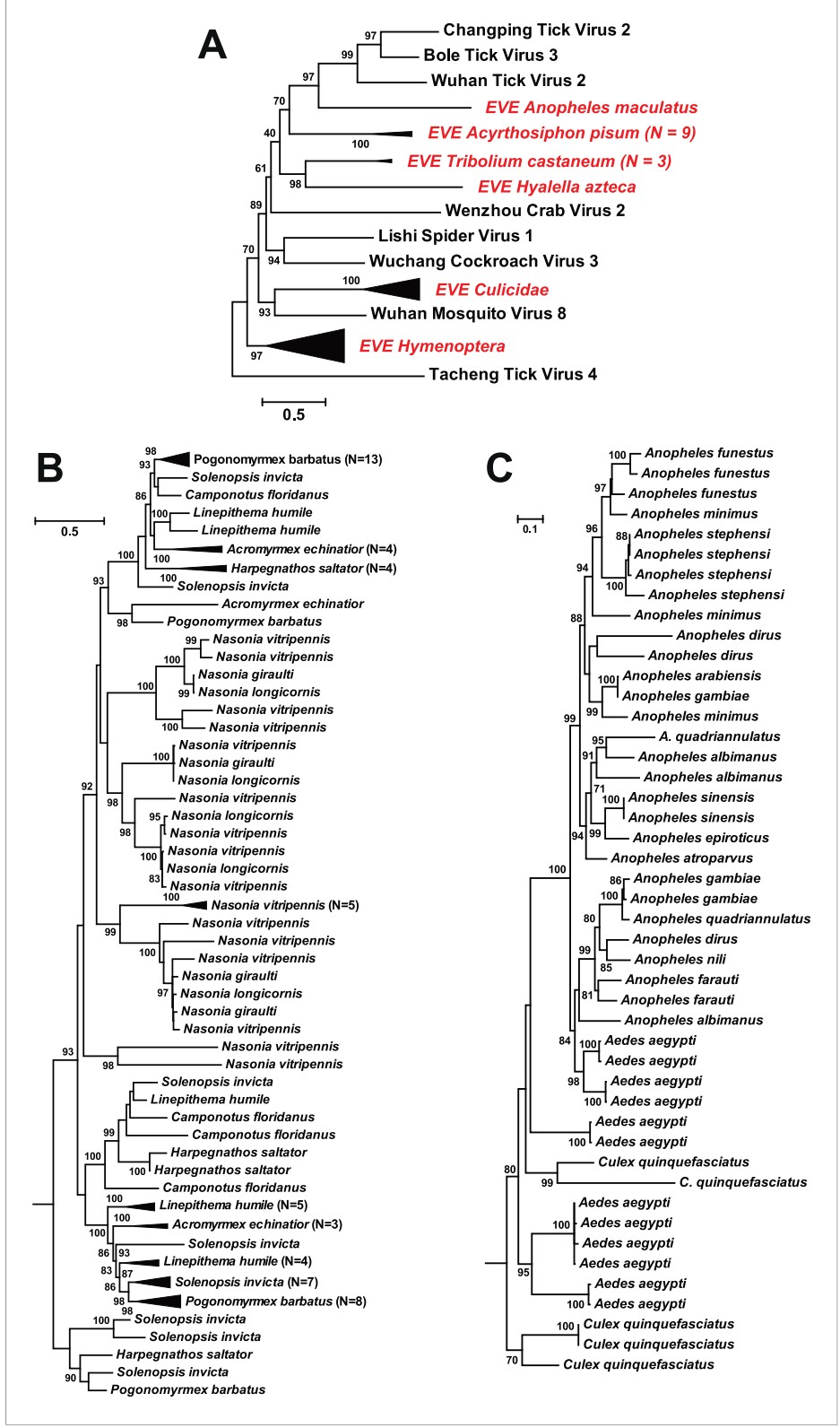

**Figure 11**. ML phylogeny of EVEs. The phylogeny is based on the glycoprotein of chuviruses in the context of exogenous members of this family (**A**), with subtrees magnified for (**B**) the Culicidae clade and (**C**) the Hymenoptera clade. The EVEs used in the phylogeny covered the complete or near complete length of the glycoprotein gene and

*Figure 11. continued on next page*

*Figure 11. Continued*
are shown in red and labeled according to host taxonomy in the overall tree. For clarity, monophyletic groups are collapsed based on the host taxonomy. Only bootstrap values >70% are shown.

using bowtie2 (*Langmead and Salzberg, 2012*). All virus genome sequences generated in this study have been deposited in the GenBank database under accession numbers KM817593–KM817764.

## Quantification of relative transcript abundances

Before quantification, we first removed the rRNA reads from the data sets to prevent any bias due to the unequal efficiency of rRNA removal steps during library preparation. To achieve this, we blasted the Trinity assembly results against the SILVER rRNA database (*Quast et al., 2013*) and then used the resulting rRNA contigs as a template for mapping using BOWTIE2 (*Langmead and Salzberg, 2012*). The remaining reads from each library were then mapped on to the assembled transcripts and analyzed with RSEM (*Li et al., 2010*), using the run_RSEM_align_n_estimate.pl scripts implemented in the Trinity program (*Grabherr et al., 2011*). The relative abundance of each transcript is presented as transcripts per million (TPM) which corrects for the total number of reads as well as for transcript length (*Li et al., 2010*).

## Genome walking

Some of the sequences obtained were substantially shorter than expected. To obtain longer sequences, we used a Genome walking kit (TaKaRa, Japan). Briefly, three gene-specific primers close to the end of the known sequence were designed. RNA from positive samples was used as input for reverse transcription primed by random primer N6. TAIL-PCR (thermal asymmetric interlaced PCR) was performed according to the manufacturer's protocol. The cDNA was used as a template for PCR with specific primers and the manufacturer-supplied degenerate primers. After three rounds of amplification, the products were analyzed on 1.0% agarose gels, and single fragments were recovered from the gels and purified using an agarose gel DNA extraction kit (TaKaRa). The purified products were then ligated into pMD19-T vector (TaKaRa) which contains the gene for ampicillin resistance. The vector was transformed into DH5α cells, which were spread on agar plates and incubated overnight at 37°C. A total of 10 clones were randomly selected and sequenced using M13 primers on ABI 3730 genetic analyzer (Applied Biosystems, Carlsbad, CA).

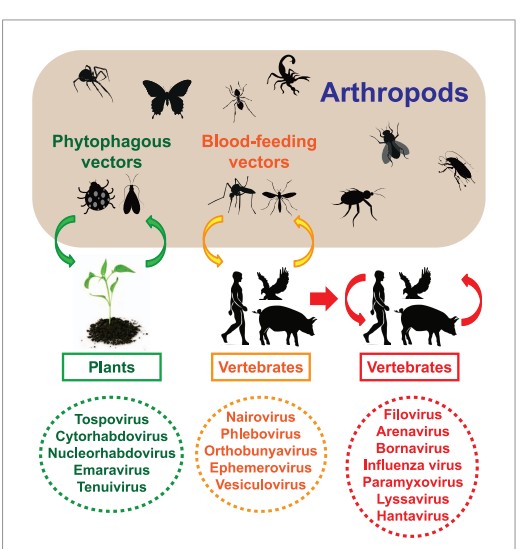

**Figure 12**. Transmission of negative-sense RNA viruses in arthropods and non-arthropods. Three types of transmission cycle are shown: (i) those between arthropods and plants are shaded green; (ii) those between arthropods and vertebrates are shaded yellow; and (iii) those that are vertebrate-only are shaded red. Viruses associated with each transmission type are also indicated.

## Determination of genome/segment termini

The extreme 5′ sequences were recovered by performing a 5′-Full RACE kit with TAP (TaKaRa) according to the manufacturer's protocol. Briefly, two gene-specific primers close to the end of the known sequence were designed. The 5′ end of RNA was ligated to the 5′RACE adaptor (without 5′ end dephosphorylating and decapping) and then reverse-transcribed using random 9 mers. The resulting cDNA was used as a template for nested PCR with 5′ RACE primers provided by the kit and gene-specific reverse primers. The PCR products were separated on an agarose gel, cloned into pMD19-T cloning vector, and subsequently sequenced.

The extreme 3′ sequences were recovered by performing a 3′-full RACE Core Set with PrimeScript RTase (TaKaRa) according to the manufacturer's protocols. Because the RNA template lacks a polyadenylated tail, a Poly(A) Tailing Kit (Applied Biosystems) was used to add this to the RNAs prior to first-strand 3′-cDNA synthesis. 20 μl of the Poly(A)-tailing reaction mixture was prepared according to the manufacturer's instructions and was incubated at 37°C for 1 hr before reverse transcription using PrimeScript Reverse Transcriptase. The cDNA was then amplified by nested PCR using the 3′ RACE primers provided by the kit and gene-specific reverse primers. The PCR products were separated on agarose gels, cloned into pMD19-T cloning vector, and subsequently sequenced. The 5′ and 3′ ends of the genome fragment were also determined by RNA circularization. RT-PCR amplification was performed across the ligated termini and the resulting PCR products were subsequently cloned and sequenced.

## Phylogenetic analyses

Potential viral proteins identified from this study were aligned with their corresponding homologs of reference negative-sense RNA viruses using MAFFT version 7 and employing the E-INS-i algorithm (*Katoh and Standley, 2013*). The sequence alignment was limited to conserved domains, with ambiguously aligned regions removed using TrimAl (*Capella-Gutierrez et al., 2009*). The final alignment lengths were 224 amino acids (aa), 412aa, 727aa, and 364aa for data sets of overall, bunya-arena-like, mononega-like, and orthomyxo-like data sets, respectively. Phylogenetic trees were inferred using the maximum likelihood method (ML) implemented in PhyML version 3.0 (*Guindon and Gascuel, 2003*), with the WAG + Γ amino acid substitution model and a Subtree Pruning and Regrafting (SPR) topology searching algorithm. Phylogenetic trees were also inferred using a Bayesian method implemented in MrBayes version 3.2.2 (*Ronquist and Huelsenbeck, 2003*), with the same substitution model as used in ML tree inference. In the MrBayes analyses, we used two simultaneous runs of Markov chain Monte Carlo sampling, and the runs were terminated upon convergence (standard deviation of the split frequencies <0.01). The phylogeny was subsequently summarized from both runs with an initial 10% of trees discarded as burn-in.

## Prediction of protein domains and functions

For each of the putative viral protein sequences, we used TMHMM v2.0 (http://www.cbs.dtu.dk/services/TMHMM/) to predict the transmembrane domains, SignalP v4.0 (http://www.cbs.dtu.dk/services/SignalP/) to determine signal sequences, and NetNGlyc v1.0 (http://www.cbs.dtu.dk/services/NetNGlyc/) to identify N-linked glycosylation sites. For some of the highly divergent viruses belonging to the Mononegavirales and the Chuviridae, a protein was regarded as a potential glycoprotein if it contained (i) a N-terminal signal domain, (ii) a C-terminal transmembrane domain, and (iii) glycosylation sites in cytoplasmic domains.

## Identification and characterization of endogenous viruses

Endogenous copies of the exogenous negative-sense RNA viruses newly described here were detected using the tBlastn algorithm against arthropod genomes available in the Reference Genomic Sequences Database (refseq_genomic) and Whole Genome Shotgun Database (WGS) in GenBank, and using viral amino acid sequences as queries. The threshold for match was set to 1e$^{-05}$ for the e-value and 50 amino acids for matched length. The query process was reversed for each potential endogenous virus to determine their corresponding phylogenetic group. Orthologous insertion events were determined by examining flanking gene sequences. Sequence alignment and phylogenetic analyses were carried out as described above.

## Characterization of bi-segmented viruses in the Chuviridae

Within the Chuviridae, Wuhan Louse Fly Virus 6 and 7, Wenzhou Crab Virus 2, Lishi Spider Virus 1, and Wuchang Cockroach Virus 3 possessed bi-segmented genomes.

Both segments were discovered using Blastx against pools of predicted proteins from unsegmented chuvirus or mononegavirales sequences. To determine that these sequences were indeed from separate segments, we performed all combinations of head-to-tail RT-PCR which allowed us to ascertain whether the sequence fragments came from a single genome. Furthermore, checking sequencing depth can help to eliminate the possibility of separate contigs being generated due to inadequate sequencing coverage. To prove that a pair of segments

belonged to the same virus, we checked: (i) sequencing depth for both segments, (ii) the presence of conserved regulatory sequences at non-coding regions of the genome, (iii) whether there is match for PCR-positive units, and (iv) the phylogenetic positions of the different viral proteins (*Figure 7A*).

## Characterization of a circular genome form within the Chuviridae

The circular genome organization within the Chuviridae was identified after we found that their genome sequences were 'over assembled' (i.e., generating contigs that contained more than one genome connected head-to-tail). This circular genomic form was also observed in both segments of the segmented chuviruses (*Figure 7B*). In addition, RT-PCR and sequencing over the entire genome did not reveal any break-points. As a control, the same protocol failed to connect the genome termini within the Mononegavirales, suggesting the circular genomic form is unique to the chuviruses. To further validate that these genomes are circular, we mapped the high-throughput sequencing reads to these assembled genomes. The coverage and depth were adequate throughout the genome with the exception of one location upstream to the 3′ end of the ORF encoding RdRp (*Figure 7C*). This genomic location had only 0–20 X coverage depending on the virus, although all RT-PCRs were successful across this location. Interestingly, sequencing of the cloned PCR products revealed extensive sequence variation (i.e., insertions and deletions) (*Figure 7C*), which is the likely cause of the low sequence coverage in this location. Collectively, these data provide strong evidence for circular genomes in the chuviruses, although this does not exclude the potential presence of linear genomic forms.

## Acknowledgements

This study was supported by National Natural Science Foundation of China (Grants 81290343, 81273014), the 12th Five-Year Major National Science and Technology Projects of China (2014ZX10004001-005). ECH is funded by an NHMRC Australia Fellowship (AF30). The authors sincerely thank Xiu-Nian Diao (Veterinary Station, Jiulingtuan of Wushi, Bole, Xinjiang Uygur Autonomous Region, China) and Ming-Hui Chen (Veterinary Station, Emin, Jiushi, Xinjiang Uygur Autonomous Region, China) for their assistance in sampling.

## Additional information

### Funding

| Funder | Grant reference number | Author |
| --- | --- | --- |
| National Natural Science Foundation of China (NSFC) | 81290343, 81273014 | Ci-Xiu Li, Mang Shi, Jun-Hua Tian, Xian-Dan Lin, Yan-Jun Kang, Liang-Jun Chen, Xin-Cheng Qin, Jianguo Xu, Edward C Holmes, Yong-Zhen Zhang |
| Ministry of Science and Technology of the People's Republic of China | 2014ZX10004001-005 | Yong-Zhen Zhang |
| National Health and Medical Research Council (NHMRC) | AF30 | Edward C Holmes |

The funders had no role in study design, data collection and interpretation, or the decision to submit the work for publication.

### Author contributions

C-XL, Acquisition of data, Analysis and interpretation of data; MS, Conception and design, Acquisition of data, Analysis and interpretation of data, Drafting or revising the article; J-HT, X-DL, Y-JK, L-JC, X-CQ, JX, Acquisition of data, Analysis and interpretation of data, Contributed unpublished essential data or reagents; ECH, Acquisition of data, Analysis and interpretation of data, Drafting or

revising the article; Y-ZZ, Conception and design, Acquisition of data, Analysis and interpretation of data, Drafting or revising the article, Contributed unpublished essential data or reagents

## Additional files

### Major datasets

The following datasets were generated:

| Author(s) | Year | Dataset title | Dataset ID and/or URL | Database, license, and accessibility information |
|---|---|---|---|---|
| Ci-Xiu Li, Mang Shi, Jun-Hua Tian, Xian-Dan Lin, Yan-Jun Kang, Liang-Jun Chen, Xin-Cheng Qin, Jianguo Xu, Edward C Holmes, Yong-Zhen Zhang | 2014 | Virus genome sequences | KM817593-KM817764 | Publicly available at NCBI GenBank (http://www.ncbi.nlm.nih.gov/genbank/). |
| Ci-Xiu Li, Mang Shi, Jun-Hua Tian, Xian-Dan Lin, Yan-Jun Kang, Liang-Jun Chen, Xin-Cheng Qin, Jianguo Xu, Edward C Holmes, Yong-Zhen Zhang | 2015 | Sequence Reads | SRP051790 | Publicly available at NCBI Sequence Read Archive (http://www.ncbi.nlm.nih.gov/sra/). |

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
