## [Decision Letter]

Thank you for sending your work entitled “Unprecedented genomic diversity of RNA
viruses in arthropods reveals the ancestry of negative-sense RNA viruses” for
consideration at *eLife*. Your article has been favorably evaluated by
Chris Ponting (Senior editor) and three reviewers, one of whom, Stephen Goff, is a
member of our Board of Reviewing Editors.

The Reviewing editor and the other reviewers discussed their comments before we reached
this decision, and the Reviewing editor has assembled the following comments to help you
prepare a revised submission.

This paper reports the recovery of sequences of a large number of new RNA viral genomes
extracted from arthropods from China. These new sequences have then been placed in a
phylogenetic tree of the virus family and reveal a hugely expanded branch of the
negative-sense RNA virus tree. There is interesting discussion about the evolution of
segmented and nonsegmented viral genomes, a current topic of great interest in
virology.

These data provide a significant amount of new food for thought about virus diversity,
virus evolution, and host range for this large and important group of viruses. We found
the paper well-written, informative, and highly accessible to virologists and especially
to nonvirologists. The text is terse and to-the-point. There are also surprises: there
is a new family, the so-called *Chuviridae*. The circular RNA genomes are
quite remarkable and require some new twists in RNA genome replication. This should
spark considerable new exploration of RNA polymerase functions.

Concerns to address in a resubmission:

The manuscript primarily requires some textual revisions to 'tone down'
some of its more general claims and to elaborate on some of the methodological
details.

1) The most substantive concern regarding the data is 'where is the rest of
it?'. The methods described would also identify positive sense RNA viruses and
dsRNA viruses: given this, why is the focus limited to negative sense viruses? Do the
other data tell a substantially different/additional/separable story? If they do not
(i.e. there is another manuscript coming with substantially the same story for +
ssRNA viruses) then we do not think it appropriate to split the dataset.

2) We would encourage some expanded discussion of the implications of the evolution of
segmented and nonsegmented viral genomes.

3) We would encourage discussion of the possibility that these viruses were not
infecting the presumed host, but were instead infecting any or all of shared gut
contents such as animal prey or food plants, gut microflora (yeasts?) or parasites
(notably nematodes and/or mites, single-celled eukaryotic parasites).

4) In the Introduction: To say arthropod viruses are 'characterised by'
persistent infection, vertical transmission and genomic integration is an extreme
overstatement, and is potentially misleading. These things happen, but surely there no
evidence that they are even that common, let alone 'characteristic'? It
might be arguable for 'persistent infection', but surely the other two
really only pertain to a minority? Also, these things seem to be ascribed to all
viruses, rather than the–ssRNA that are the focus of the manuscript.

5) In the Results, the authors state that “a key result of this study is that the
majority of the diversity of negative-sense RNA viruses now appears to be contained
within viruses that utilize arthropods as hosts or vectors”: this over claims the
case. This large-scale systematic metagenomic survey (relative to the small-scale ad hoc
analyses that represent the majority of the previously described–ssRNA viruses)
certainly identifies an exciting diversity; but who's to say that a similar
survey of marine molluscs (or basal metazoan?) wouldn't change the perspective
again? The only strong claim that can be made here is that viruses are undersampled.
This is revisited in the first paragraph of the Discussion section, where the authors
acknowledge the problem, but we think it needs to be toned down.

6) We don't really 'buy' the argument in the second paragraph of
the Discussion section that 'RNAi enables benign and persistent
infections'. Since vertebrates are the only deep eukaryotic lineage (probably)
lacking the antiviral RNAi response, this amounts to claiming that vertebrates are
unique is not easily developing persistent benign viral infections (what about plants
and fungi?). Is this really the case? What evidence is there to support this?

7) It seems a missed opportunity to not discuss timescales. We appreciate this is hard,
but given the large diversity of new viruses here (and the ease with which some other
RNA viruses form endogenous copies/EVEs) it seems a shame not to search genomic DNA
datasets to identify shared ancient insertions that could be used to date some of the
splits.

8) In the Results, where two different hosts are identified for a 'single'
virus, we need to know the (synonymous site and amino acid) divergence between the two
different viral lineages. The timescale over which such transmission occurs is
vital—they could be sharing an infection pool, or they might not have had a
common ancestor for thousands of years.

9) The experimental approach is quite straightforward, but comprehensive and well
conducted, and the results are impressive. But it is unclear what efforts have been
taken by the authors to ensure that the viruses were indeed from arthropods and not
internal or external contaminants. This point should at least be discussed.

10) The results are presented in a rather cursory manner, and we would like to see a lot
more detail, such as fully labeled trees being available in supplementary
information.

11) While we appreciate that the RdRp is highly conserved and unlikely to provide strong
statistical supports, there are unequivocal statements about phylogenetic relationships
that may not have any support. Sequencing of other more variable genes and a more
considered analysis would help to resolve this. As it is, it seems that only 171
sequences were generated from this study (sixth paragraph of the Materials and methods
section), which is baffling, and it is not clear whether the deep sequencing assemblies
are going to be made available anywhere (please provide details).

12) Ideally, we would like to see some attempt made at dating these new viruses but we
understand that this could be very challenging.

---

## [Author Response]

*1) The most substantive concern regarding the data is 'where is the rest
of it?'. The methods described would also identify positive sense RNA viruses
and dsRNA viruses: given this, why is the focus limited to negative sense viruses? Do
the other data tell a substantially different/additional/separable story? If they do
not (i.e. there is another manuscript coming with substantially the same story for
+ ssRNA viruses) then we do not think it appropriate to split the
dataset*.

The reviewers are correct that we did indeed discover a number of other RNA viruses
(positive-sense, double-stranded) in the arthropods examined here. However, the newly
discovered diversity in other virus groups, although novel, is not as comprehensive and
all-encompassing as that of the negative-sense RNA viruses described here, and tells a
rather different story. In particular, the positive- and double-stranded RNA viruses
(which we are still characterizing) tend to fall within the diversity of known families,
rather than filling phylogenetic ‘gaps’ as is the case with the
negative-sense RNA viruses. In addition, the role played by arthropods in such families
as the *Flaviviridae* and the *Picornavirales* is somewhat
different (and more complex) to what we present here (for example, see our recent paper
on the evolutionary and genomic complexity of the *Flaviviridae*: Qin et
al., PNAS, 111, 6744-6749, 2014). We therefore feel that it is difficult to present a
single, coherent picture for all these RNA viruses, and that some families of
positive-sense viruses will likely merit individual publications. Hence, because the
negative-sense RNA viruses are a distinct phylogenetic group, we feel that it is
simplest to present a stand-alone paper on these viruses, rather than greatly
complicating it with the inclusion of other viral groups.

*2) We would encourage some expanded discussion of the implications of the
evolution of segmented and nonsegmented viral genomes*.

We agree with the reviewers that the implications of virus segmentation could be
discussed more thoroughly and we have now added some additional text on this matter.
Because we would like to avoid unnecessary speculation, and it seems premature to make
strong claims in the absence of a wider sampling of eukaryotic viruses, we have focused
this extra text on the future work that needs to be done. The expanded discussion can be
found in the sixth paragraph of the Discussion section:

“Independent changes of genome segmentation numbers are also observed in the
mononegavirales-like viruses (Figure 4) and, more
frequently, in the chuviruses (Figure 7).
Consequently, the number of genome segments appears to be a relatively flexible trait at
a broad evolutionary scale, although the functional relevance of these changes remains
unclear. While the segmented viruses (bunya-arenaviruses, orthomyxoviruses, and
ophioviruses) appear to be distinct from the largely unsegmented mononegavirales-like
viruses in our phylogenetic analysis, this may be an artifact of under-sampling,
especially given that only a tiny fraction of eukaryotes have been sampled to date. With
a wider sample of eukaryotic viruses it will be possible to more accurately map changes
in segment number onto phylogenetic trees and in so doing come to a more complete
understanding of the patterns and determinants of the evolution of genome
segmentation.”

*3) We would encourage discussion of the possibility that these viruses were not
infecting the presumed host, but were instead infecting any or all of shared gut
contents such as animal prey or food plants, gut microflora (yeasts?) or parasites
(notably nematodes and/or mites, single-celled eukaryotic parasites)*.

The reviewers make a very good point, and we agree that the potential source of these
viruses should be examined and discussed in more detail. Although it is impossible to
achieve a clear picture of the true source for every virus in a metagenomic sequencing
project such as this, we found that by measuring the relative abundance of viral
transcripts we could at least distinguish those viruses that are clearly infecting the
host organism we have sampled (i.e. those with high abundance) from those with possible
other origins (which we predict to be of low abundance). Furthermore, the possibility
that food/microflora/parasites are the source of the viruses is further reduced if the
less abundant viruses appear closely related to the abundant viruses in phylogenetic
trees. Collectively, we believe that this approach provides an effective working
solution to the source of the viruses observed here. The description of RNA transcript
quantification can be found in the Methods section in the seventh paragraph:

“Quantification of relative transcript abundances. Before quantification, we
first removed the rRNA reads from the data sets to prevent any bias due to the unequal
efficiency of rRNA removal steps during library preparation. To achieve this, we blasted
the Trinity assembly results against the SILVER rRNA database (19), and then used the resulting rRNA contigs as a
template for mapping using BOWTIE2 (13). The remaining reads from each library were then mapped on to
the assembled transcripts and analyzed with RSEM (14), using the run_RSEM_align_n_estimate.pl scripts implemented in
the Trinity program (7). The
relative abundance of each transcript is presented as transcripts per million (TPM)
which corrects for the total number of reads as well as for transcript length (14).”

The new results include an additional figure and an extra “Abundance”
column in Tables 2, 3 and 4. The
revised text can be found in the second paragraph of the Results section and reads:

“Next, we measured the abundance of these sequences as the number transcripts per
million (TPM) within each library after the removal of rRNA reads. The abundance of
viral transcripts calculated in this manner exhibited substantial variation (Figure 2, Tables 2, 3 and 4): while the least abundant L segment
(Shayang Spider Virus 3) contributed to less than 0.001% to the total non-ribosomal RNA
content, the most abundant (Sanxia Water Strider Virus 1) was at a frequency of 21.2%,
and up to 43.9% if we include the matching M and S segments of the virus. The remaining
viral RdRp sequences fell within a range (10-1000 TPM) that matched the abundance level
of highly expressed host mitochondrial genes (Figure 2).”

Finally, in the Discussion section of the manuscript, we examined the potential causes
of this variation of abundance level, including the possibility that they did not infect
the presumed host. The revised text can be found in the third paragraph and reads as
follows:

“The viruses discovered here also exhibited a huge variation in level of
abundance. It is possible that this variation is in part due to the stage or severity of
infection in individual viruses, and may be significantly influenced by the process of
pooling, since most of our libraries contain an uneven mixture of different host species
or even genera. In addition, it is possible that some low abundance viruses may in fact
be derived from other eukaryotic organisms present in the host sampled, such as
undigested food or prey, gut micro flora, and parasites. Nevertheless, since the
majority of the low abundance viruses appear in the same groups as the highly abundant
ones in our phylogenetic analyses, these viruses are most likely associated with
arthropods.”

*4) In the Introduction: To say arthropod viruses are 'characterised
by' persistent infection, vertical transmission and genomic integration is an
extreme overstatement, and is potentially misleading. These things happen, but surely
there no evidence that they are even that common, let alone
'characteristic'? It might be arguable for 'persistent
infection', but surely the other two really only pertain to a minority? Also,
these things seem to be ascribed to all viruses, rather than the–ssRNA that
are the focus of the manuscript*.

We agree with the reviewers that this was an overstatement and the relevant text has
therefore been deleted.

*5) In the Results, the authors state that “a key result of this study is
that the majority of the diversity of negative-sense RNA viruses now appears to be
contained within viruses that utilize arthropods as hosts or vectors”: this
over claims the case. This large-scale systematic metagenomic survey (relative to the
small-scale* ad hoc *analyses that represent the majority of the
previously described–ssRNA viruses) certainly identifies an exciting
diversity; but who's to say that a similar survey of marine molluscs (or basal
metazoan?) wouldn't change the perspective again? The only strong claim
that* can *be made here is that viruses are undersampled. This is
revisited in the first paragraph of the Discussion section, where the authors
acknowledge the problem, but we think it needs to be toned down*.

We apologize for the confusion caused by our wording, which we agree was too strong. The
intended point was simply that currently known virus diversity (which is largely from
vertebrates and plants) is mainly nested within the viral diversity newly discovered in
arthropods. The relevant context has therefore been changed in the fourth paragraph of
the Results section:

“A key result of this study is that the much of the genetic diversity of
negative-sense RNA viruses in vertebrates and plants now appears to be contained within
viruses that utilize arthropods as hosts or vectors.”

And also in the first paragraph of the Discussion section:

“Our study suggests that arthropods are major reservoir hosts for many, if not
all, of the negative-sense RNA viruses in vertebrates and plants, and hence have likely
played a major role in their evolution.”

*6) We don't really 'buy' the argument in the second
paragraph of the Discussion section that 'RNAi enables benign and persistent
infections'. Since vertebrates are the only deep eukaryotic lineage (probably)
lacking the antiviral RNAi response, this amounts to claiming that vertebrates are
unique is not easily developing persistent benign viral infections (what about plants
and fungi?). Is this really the case? What evidence is there to support
this*?

The reviewers make a fair point. Accordingly, the relevant text has been deleted, and
replaced by the following text in the second paragraph of the Discussion section:

“Furthermore, arthropods are involved in almost all ecological guilds and
actively interact with other eukaryotes, including animals, plants and fungi, such that
it is possible that they serve as both sources and sinks for viruses present in the
environment. In addition, not only were diverse viruses present, but they were often
highly abundant. For example, in the pool containing twelve individuals (representing
two species) from the *Gerridae* (Water striders) collected at the same
site, we identified at least five negative-sense RNA viruses whose TPM values are well
above 100, and where the viral RNA collectively made up more than 50% of the host total
RNA (rRNA excluded). Determining why arthropods are able to carry such a large viral
diversity and at such frequencies clearly merits further investigation.”

*7) It seems a missed opportunity to not discuss timescales. We appreciate this
is hard, but given the large diversity of new viruses here (and the ease with which
some other RNA viruses form endogenous copies/EVEs) it seems a shame not to search
genomic DNA datasets to identify shared ancient insertions that could be used to date
some of the splits*.

As the reviewers correctly state, inferring the time-scale of this evolutionary history
is very difficult (although it is clearly ancient) and something we considered closely
(because it is obviously very interesting). We did find a large number of potential EVEs
using tBlastx (please see the new Table 5 in
the paper). Interestingly, most of these EVE sequences have disrupted reading frames and
many are found within transposon elements, suggesting that the transposons have been
central to their integration. Nevertheless, and critically for the context of molecular
clock dating, the EVEs we discovered do not share homologous integration sites in the
host genome as revealed by an analysis of flanking sequences (i.e. they are likely not
orthologous and orthology greatly aids dating). In addition, many are in short
contig/scaffolds such that it is difficult to trace their locations. Furthermore,
phylogenetic analyses of those EVEs shared among different host species revealed
extremely complex tree topologies which do not reflect the host phylogeny; these are
indicative of independent integration events that again prohibit effective molecular
clock dating (please find an example tree in the new Figure 11). In sum, after a careful analysis of the EVE data, we could not
find any clear examples that could provide a reliable time-scale of viral evolution. In
addition, the general incongruence between the (exogenous) virus and host phylogenies
precluded any reliable molecular clock dating analysis. We would therefore prefer not to
include any such analysis as it will clearly only add error (i.e. we cannot safely say
that these dating estimates will be accurate). These points are now made in the revised
version of the paper. In addition, we have added some text outlining the discovery and
characterization of the novel endogenous viruses. The description of the methods can be
found in the thirteenth paragraph of the Materials and methods section and reads:

“Identification and characterization of endogenous viruses. Endogenous copies of
the exogenous negative-sense RNA viruses newly described here were detected using the
tBlastn algorithm against arthropod genomes available in the Reference Genomic Sequences
Database (refseq_genomic) and Whole Genome Shotgun Database (WGS) in GenBank, and using
viral amino acid sequences as queries. The threshold for match was set to 1e-05 for the
e-value and 50 amino acids for matched length. The query process was reversed for each
potential endogenous virus to determine their corresponding phylogenetic group.
Orthologous insertion events were determined by examining flanking gene sequences.
Sequence alignment and phylogenetic analyses were carried out as described
above.”

These results are described in a new Figure 11
and Table 5. The related text can be found in
the last paragraph of the Results section and reads:

“Novel Endogenous Virus Elements (EVEs). As well as novel exogenous RNA viruses,
our metagenomic analysis also revealed a large number of potential EVEs (11) in more than 40
arthropod species: these resembled complete or partial genes of the major
proteins—the nucleoprotein, glycoprotein and RdRp—but without fully intact
genomes (Table 5). As expected given their
endogenous status, most of these sequences have disrupted reading frames and many are
found within transposon elements, suggesting that transposons have been central to their
integration. Interestingly, in some cases, such as the putative glycoprotein gene of
chuviruses, the homologous EVEs from within a family (*Culicidae*) or
even an order (*Hymenoptera*) form monophyletic groups (Figure 11). However, they are unlikely to be
orthologous because they do not share homologous integration sites in the host genome as
determined by an analysis of flanking sequences, which in turn limited the applicability
of molecular-clock based dating techniques. Furthermore, phylogenetic analyses of those
EVEs shared among different host species revealed extremely complex tree topologies
which do not exhibit simple matches to the host phylogeny at both the species and genera
levels (Figure 11). In sum, these results
suggest that EVEs are relative commonplace in arthropod genomes and have been often
generated by multiple and independent integration events.”

*8) In the Results, where two different hosts are identified for a
'single' virus, we need to know the (synonymous site and amino acid)
divergence between the two different viral lineages. The timescale over which such
transmission occurs is vital—they could be sharing an infection pool, or they
might not have had a common ancestor for thousands of years*.

The genetic diversity of these viruses is very limited across different hosts and the
viral phylogenies do not reflect those of their hosts. As a consequence, it seems
reasonable to conclude that they are likely from the same infection pool. We have
included a new figure illustrating the relationship between these viruses in the context
of multiple hosts, and the revised text can be found in the fourth paragraph of the
Results section:

“Interestingly, host species in the same niche had similar viral contents that
were generally incongruent with the host phylogeny (Figure 6). Such a pattern is indicative of frequent cross-species and
occasional cross-genus virus transmission in the context of ecological and geographic
proximity.”

*9) The experimental approach is quite straightforward, but comprehensive and
well conducted, and the results are impressive. But it is unclear what efforts have
been taken by the authors to ensure that the viruses were indeed from arthropods and
not internal or external contaminants. This point should at least be
discussed*.

Please refer to our response to Major Comment #3.

*10) The results are presented in a rather cursory manner, and we would like to
see a lot more detail, such as fully labeled trees being available in supplementary
information*.

We apologize for the over-simplification of the Results; we were trying to be as
succinct as possible. Detailed phylogenic trees have now been included as Figure 3—figure supplement 3.

*11) While we appreciate that the RdRp is highly conserved and unlikely to
provide strong statistical supports, there are unequivocal statements about
phylogenetic relationships that may not have any support. Sequencing of other more
variable genes and a more considered analysis would help to resolve this. As it is,
it seems that only 171 sequences were generated from this study (sixth paragraph of
the Materials and methods section), which is baffling, and it is not clear whether
the deep sequencing assemblies are going to be made available anywhere (please
provide details)*.

We agree with the reviewers that the RdRp phylogeny may lack resolution if: (i) the
sequences are too conserved (i.e. below species level), and (ii) the sequence are too
divergent such that they do not share conserved domains. However, in our case, the
replicase (RdRp) domains were well conserved across the negative-sense RNA viruses such
that a reliable topology could be inferred (and very similar phylogenies were obtained
using both ML and Bayesian methods). In contrast, the other major viral genes
(glycoproteins and nucleoprotein) are far more divergent and lack conserved domains
across genera (some are not even homologous), and therefore they are not helpful in a
phylogenetic analysis aimed at resolving ancient relationships. As the result, the only
viable way to infer the deep phylogenetic relationships among negative-sense viruses is
with the RdRp which, of course, is why there is a rich literature on the use of this
gene but not any others. We therefore strongly believe that our phylogenetic analysis of
RdRp is the only scientifically rigorous way to proceed.

To clarify, the 171 sequences we deposited in the GenBank are confirmed complete or near
complete genomes of all the negative-sense RNA viruses found in this study. We agree
with the reviewers that raw data should be made available and, consequently, all the
reads generated in this study have now been uploaded to NCBI Sequence Read Achieve. The
corresponding description is added in the fourth paragraph of the Materials and methods
section:

“All sequence reads generated in this study were uploaded onto NCBI Sequence Read
Achieve (SRA) database under the BioProject accession SRP051790.”

*12) Ideally, we would like to see some attempt made at dating these new viruses
but we understand that this could be very challenging*.

Please refer to our response to Major Comment #7. Sadly, we believe that dating
is too difficult to perform on these data and will just add error.